# Distributional Reinforcement Learning with Regularized Wasserstein Loss

**Ke Sun[1], Yingnan Zhao[2], Wulong Liu[3], Bei Jiang[1], Linglong Kong[1]***
[1]University of Alberta, Edmonton, Canada
[2] Harbin Engineering University, China
[3]Huawei Noah's Ark Lab
{ksun6,bei1,lkong}@ualberta.ca
zhaoyingnan@hrbeu.edu.cn
liuwulong@huawei.com

## Abstract

The empirical success of distributional reinforcement learning (RL) highly relies on the choice of distribution divergence equipped with an appropriate distribution representation. In this paper, we propose *Sinkhorn distributional RL (Sinkhorn-DRL)*, which leverages Sinkhorn divergence—a regularized Wasserstein loss—to minimize the difference between current and target Bellman return distributions. Theoretically, we prove the contraction properties of SinkhornDRL, aligning with the interpolation nature of Sinkhorn divergence between Wasserstein distance and Maximum Mean Discrepancy (MMD). The introduced SinkhornDRL enriches the family of distributional RL algorithms, contributing to interpreting the algorithm behaviors compared with existing approaches by our investigation into their relationships. Empirically, we show that SinkhornDRL consistently outperforms or matches existing algorithms on the Atari games suite and particularly stands out in the multi-dimensional reward setting. Code is available in https://github.com/datake/SinkhornDistRL..

## 1 Introduction

The design of classical reinforcement learning (RL) algorithms primarily focuses on the expectation of cumulative rewards that an agent observes while interacting with the environment. Recently, a new class of RL algorithms called *distributional RL* estimates the full distribution of total returns and has exhibited state-of-the-art performance in a wide range of environments, such as C51 [5], Quantile-Regression DQN (QR-DQN) [14], EDRL [41], Implicit Quantile Networks (IQN) [13], Fully Parameterized Quantile Function (FQF) [56], Non-Crossing QR-DQN [58], Maximum Mean Discrepancy (MMD-DQN) [37], Spline (SPL-DQN) [33], and Sketch-DQN [53]. Beyond the performance advantage, distributional RL has also possessed benefits in risk-sensitive control [13, 9], exploration [35, 11, 48], offline setting [34, 55], statistical value estimation [43], robustness [47] and optimization [46, 52, 42, 51].

**Limitations of Typical Distributional RL Algorithms.** Despite the gradual introduction of numerous algorithms, quantile regression-based algorithms [14, 13, 56, 41, 33, 42, 43] dominate attention and research in the realm of distributional RL. These algorithms utilize quantile regression to approximate the one-dimensional Wasserstein distance to compare two return distributions. Nevertheless, two major limitations hinder their performance improvement and wider practical deployment. *1) Inaccuracy in Capturing Return Distribution Characteristics.* The way of directly

38th Conference on Neural Information Processing Systems (NeurIPS 2024).

---

*Corresponding author

generating quantiles of return distributions via neural networks often suffers from the non-crossing issue [58], where the learned quantile curves fail to guarantee a non-decreasing property. This leads to abnormal distribution estimates and reduced model interpretability. The inaccurate distribution estimate is fundamentally attributed to the use of pre-specified statistics [41], while unrestricted statistics based on deterministic samples can be potentially more effective in complex environments [37]. *2) Difficulties in Extension to Multi-dimensional Rewards.* Many RL tasks involve multiple sources of rewards [32, 15], hybrid reward architecture [50, 30], or sub-reward structures after reward decomposition [31, 57], which require learning multi-dimensional return distributions to reduce the intrinsic uncertainty of the environments. However, it remains elusive how to use quantile regressions to approximate a multi-dimensional Wasserstein distance, while circumventing the computational intractability issue in the related multi-dimensional output space.

**Motivation of Sinkhorn Divergence: a Regularized Wasserstein loss.** Sinkhorn divergence [45] has emerged as a theoretically principled and computationally efficient alternative for approximating Wasserstein distance. It has gained increasing attention in the field of optimal transport [4, 24, 21, 39] and has been successfully applied in various areas of machine learning [38, 25, 54, 20, 8]. By introducing entropic regularization, Sinkhorn divergence can efficiently approximate a multi-dimensional Wasserstein distance using computationally efficient matrix scaling algorithms [45, 39]. This makes it feasible to apply optimal transport distances to RL tasks with multi-dimensional rewards (see experiments in Section 5.3). Moreover, Sinkhorn divergence enables the leverage of samples to approximate return distributions instead of relying on pre-specified statistics, e.g., quantiles, thereby increasing the accuracy in capturing the full data complexity behind return distributions and naturally avoiding the non-crossing issues in distributional RL. Beyond addressing the two main limitations mentioned above, the well-controlled regularization introduced in Sinkrhorn divergence helps to find a "smoother" transport plan relative to Wasserstein distance, making it less sensitive to noises or small perturbations when comparing two return distributions (see Appendix A for the visualization). The term "smoother" refers to the effect of regularization in Sinkhorn divergence to encourage a more uniformly distributed transport plan. This regularization also aligns with the maximum-entropy principle [28, 16], which aims to maximize entropy while keeping the transportation cost constrained. Furthermore, the resulting strongly convex loss function [3] and the induced smoothness by regularization facilitate faster and more stable convergence in the deep RL setting (see more details in Sections 4 and 5).

**Contributions.** In this work, we propose a new family of distributional RL algorithms based on Sinkhorn divergence, a regularized Wasserstein loss, to address the limitations of quantile regression-based algorithms while promoting more stable training. As Sinkhorn divergence interpolates between Wasserstein distance and MMD [26, 21, 39], we probe this relationship in the RL context, characterizing the convergence properties of dynamic programming under Sinkhorn divergence and revealing the connections of different distances. Our study enriches the class of distributional RL algorithms, making them more effective for a broader range of scenarios and potentially inspiring advancement in other related areas of distribution learning. Our key contributions are summarized as follows:

**(1) Algorithm.** We introduce a Sinkhorn distributional RL algorithm, called SinkhornDRL, which overcomes the primary shortcomings of predominantly utilized quantile regression-based algorithms. SinkhornDRL can be seamlessly integrated into existing model architectures and easily implemented.

**(2) Theory.** We establish the properties of Sinkhorn divergence within distributional RL and derive the relevant convergence results for (multi-dimensional) distributional dynamic programming.

**(3) Experiments.** We conduct an extensive comparison of SinkhornDRL with typical distributional RL algorithms across 55 Atari games, performing rigorous sensitivity analyses and computation cost assessments. We also verify the efficacy of SinkhornDRL in the multi-dimensional reward setting.

## 2  Preliminary Knowledge

### 2.1  Distributional Reinforcement Learning

In classical RL, an agent interacts with an environment via a Markov decision process (MDP), a 5-tuple $(\mathcal{S}, \mathcal{A}, R, P, \gamma)$, where $\mathcal{S}$ and $\mathcal{A}$ are the state and action spaces, $P$ is the environment transition dynamics, $R$ is the reward function, and $\gamma \in (0, 1)$ is the discount factor.

Given a policy $\pi$, the discounted sum of future rewards $Z^\pi$ is a random variable with $Z^\pi(s, a) = \sum_{t=0}^\infty \gamma^t R(s_t, a_t)$, where $s_0 = s$, $a_0 = a$, $s_{t+1} \sim P(\cdot|s_t, a_t)$, and $a_t \sim \pi(\cdot|s_t)$. In expectation-based RL, the action-value function $Q^\pi$ is defined as $Q^\pi(s, a) = \mathbb{E}\left[Z^\pi(s, a)\right]$, which is iteratively updated via Bellman operator $\mathcal{T}^\pi$ through $\mathcal{T}^\pi Q(s, a) = \mathbb{E}[R(s, a)] + \gamma \mathbb{E}_{s' \sim p, \pi}\left[Q\left(s', a'\right)\right]$, where $s' \sim P(\cdot|s, a)$ and $a' \sim \pi(\cdot|s')$. In contrast, distributional RL focuses on the action-return distribution, the full distribution of $Z^\pi(s, a)$. The return distribution is iteratively updated by applying the distributional Bellman operator $\mathfrak{T}^\pi$ through $\mathfrak{T}^\pi Z(s, a) :\overset{D}{=} R(s, a) + \gamma Z\left(s', a'\right)$, where $D$ denotes the distribution and the equality implies random variables of both sides are equal in distribution. The distributional Bellman operator $\mathfrak{T}^\pi$ is contractive under certain distribution divergence metrics [19].

## 2.2 Divergences between Measures

**Optimal Transport (OT) and Wasserstein / Earth Mover's Distance.** The optimal transport (OT) metric $W_c$ defines a powerful geometry to compare two probability measures $(\mu, \nu)$, i.e., $W_c = \inf_{\Pi \in \mathbf{\Pi}(\mu, \nu)} \int c(x, y) \mathrm{d}\Pi(x, y)$, where $c$ is the cost function, $\Pi$ is the joint distribution with marginals $(\mu, \nu)$, and the minimizer $\Pi^*$ is called the *optimal transport plan* or *optimal coupling*. The $p$-Wasserstein distance $W_p = (\inf_{\Pi \in \mathbf{\Pi}(\mu, \nu)} \int \|x - y\|^p \mathrm{d}\Pi(x, y))^{1/p}$ is a special case of optimal transport with the Euclidean norm as the cost function. Relative to conventional divergences, including Hellinger, total variation or Kullback-Leibler divergences, the formulation of OT and Wasserstein distance inherently integrates the spatial or geometric relationships between data points and allows them to recover the full support of measures. This theoretical advantage comes, however, with a heavy computational price tag, especially in the high-dimensional space. Specifically, finding the optimal transport plan amounts to solving a linear program and the cost scales at least in $\mathcal{O}(d^3 \log(d))$ when comparing two histograms of dimension $d$ [12].

**Maximum Mean Discrepancy [26].** Define two random variables $X$ and $Y$. The squared Maximum Mean Discrepancy (MMD) $\mathrm{MMD}_k^2$ with the kernel $k$ is formulated as $\mathrm{MMD}_k^2 = \mathbb{E}\left[k\left(X, X'\right)\right] + \mathbb{E}\left[k\left(Y, Y'\right)\right] - 2\mathbb{E}\left[k(X, Y)\right]$, where $k(\cdot, \cdot)$ is a continuous kernel and $X'$ (resp. $Y'$) is a random variable independent of $X$ (resp. $Y$). Mathematically, the "flat" geometry that MMD induces on the space of probability measures does not faithfully lift the ground distance [21], potentially inferior to OT when comparing two complicated distributions. However, MMD is cheaper to compute than OT with a smaller *sample complexity*, i.e., the number of samples for measures to approximate the true distance [24]. We provide more details of various distribution divergences as well as their existing contraction properties in distributional RL in Appendix B.

**Notations.** We constantly use the *unrectified kernel* $k_\alpha = -\|x - y\|^\alpha$ in our algorithm analysis. With a slight abuse of notation, we also use $Z_\theta$ to denote $\theta$ parameterized return distribution.

## 3  Related Work

Based on the choice of distribution divergences and the distribution representation, distributional RL algorithms can be classified into three categories.

**(1) Categorical Distributional RL.** As the first successful class, categorical distributional RL [5], e.g., C51, represents the return distribution using a categorical distribution with discrete fixed supports within a predefined interval.

**(2) Quantile Regression (Wasserstein Distance) Distributional RL.** QR-DQN [14] employs quantile regression to approximate the one-dimensional Wasserstein distance. It learns the quantile values for a series of fixed quantiles, offering greater flexibility in the support compared with categorical distributional RL. IQN [13] enhances this approach by utilizing an implicit model to produce more expressive quantile values, instead of fixed ones in QR-DQN, while FQF [56] further advances IQN by introducing a more expressive quantile network. However, as mentioned in Section 1, quantile regression distributional RL struggles with accurately capturing return distribution characteristics and handling multi-dimensional reward settings. SinkhornDRL, with the assistance of an entropy regularization, offers an alternative approach that addresses the two limitations simultaneously.

**(3) MMD Distributional RL.** Rooted in kernel methods [26, 53], MMD-DQN [37] learns unrestricted statistics, i.e., samples, to represent the return distribution and optimizes under MMD, which can manage multi-dimensional rewards. However, the data geometry captured by MMD with a specific

kernel may be limited, as it is highly sensitive to the characteristics of kernels and the induced Reproducing Kernel Hilbert space (RKHS) [25, 26, 23]. In contrast, SinkhornDRL is fundamentally based on OT, inherently capturing the spatial and geometric layout of return distributions. This enables SinkhornDRL to potentially surpass MMD-DQN by leveraging a richer representation of data geometry. In Section 5, we present extensive experiments to demonstrate the advantage of SinkhornDRL over MMD-DQN, particularly in the multi-dimensional reward scenario in Section 5.3.

## 4 Sinkhorn Distributional RL (SinkhornDRL)

The algorithmic evolution of distributional RL can be primarily viewed along two dimensions [37]. (1) Introducing new distributional RL families beyond the three established ones, leveraging alternative distribution divergences combined with suitable density estimation techniques. (2) Enhancing existing algorithms within a particular family by increasing their model capacity, e.g., IQN and FQF. Concretely, SinkhornDRL falls into the first dimension, aiming to expand the range of distributional RL algorithm families.

### 4.1 Sinkhorn Divergence and New Convergence Properties in Distributional RL

Sinkhorn divergence [45] efficiently approximates the optimal transport problem by introducing an entropic regularization. It aims at finding a sweet trade-off that simultaneously leverages the geometry property of Wasserstein distance (optimal transport distances) and the favorable sample complexity advantage and unbiased gradient estimates of MMD [25, 21]. For two probability measures $\mu$ and $\nu$, the entropic regularized Wasserstein distance $\mathcal{W}_{c,\varepsilon}(\mu, \nu)$ is formulated as

$$\mathcal{W}_{c,\varepsilon}(\mu, \nu) = \min_{\Pi \in \mathbf{\Pi}(\mu, \nu)} \int c(x, y) \mathrm{d}\Pi(x, y) + \varepsilon \mathrm{KL}(\Pi | \mu \otimes \nu), \tag{1}$$

where the entropic regularization $\mathrm{KL}(\Pi | \mu \otimes \nu) = \int \log \left( \frac{\Pi(x,y)}{\mathrm{d}\mu(x)\mathrm{d}\nu(y)} \right) \mathrm{d}\Pi(x, y)$, also known as *mutual information*, makes the optimization strongly convex and differential [3, 21], allowing for efficient matrix scaling algorithms for approximation, such as Sinkhorn Iterations [45]. In statistical physics, $\mathcal{W}_{c,\varepsilon}(\mu, \nu)$ can be re-factored as a projection problem:

$$\mathcal{W}_{c,\varepsilon}(\mu, \nu) := \min_{\Pi \in \mathbf{\Pi}(\mu, \nu)} \mathrm{KL}\left(\Pi | \mathcal{K}\right), \tag{2}$$

where $\mathcal{K}$ is the Gibbs distribution and its density function satisfies $d\mathcal{K}(x, y) = e^{-c(x,y)/\varepsilon} d\mu(x) d\nu(y)$. This problem is often referred to as the "static Schrödinger problem" [29, 44] as it was initially considered in statistical physics. Formally, the Sinkhorn divergence is defined as

$$\overline{\mathcal{W}}_{c,\varepsilon}(\mu, \nu) = 2\mathcal{W}_{c,\varepsilon}(\mu, \nu) - \mathcal{W}_{c,\varepsilon}(\mu, \mu) - \mathcal{W}_{c,\varepsilon}(\nu, \nu), \tag{3}$$

which is smooth, positive definite, and metricizes the convergence in law [21]. This definition subtracts two self-distance terms to ensure non-negativity and metric properties.

**Properties for Convergence.** The contraction analysis of distributional Bellman operator $\mathfrak{T}^\pi$ under a distribution divergence $d_p$ depends on its *scale sensitive* (**S**) and *sum invariant* (**I**) properties [6, 5]. We say $d_p$ is scale sensitive (of order $\tau$) if there exists a $\tau > 0$, such that for all random variables $X, Y$ and a real value $a > 0$, $d_p(aX, aY) \le |a|^\tau d_p(X, Y)$. $d_p$ has the sum invariant property if whenever a random variable $A$ is independent from $X, Y$, we have $d_p(A + X, A + Y) \le d_p(X, Y)$. Based on these properties, [5] shows that $\mathfrak{T}^\pi$ is $\gamma$-contractive under the supremal form of Wasserstein distance $W_p$, which is regarding the first term of $\mathcal{W}_{c,\varepsilon}$ or directly letting $\varepsilon = 0$ in Eq. 1. When examining the regularized loss form of $\mathcal{W}_{c,\varepsilon}$, a natural question arises: *What is the influence of the incorporated regularization term on the contraction of $\mathfrak{T}^\pi$?* We begin to address this question in Proposition 1, focusing on the separate regularization term. Here, we define mutual information as $\mathrm{MI}_\Pi(\mu(s, a), \nu(s, a)) = \mathrm{KL}(\Pi | \mu(s, a) \otimes \nu(s, a))$ and its supremal form $\mathrm{MI}_\Pi^\infty(\mu, \nu) = \sup_{(s,a) \in \mathcal{S} \times \mathcal{A}} \mathrm{KL}(\Pi | \mu(s, a) \otimes \nu(s, a))$ given a joint distribution $\Pi$.

**Proposition 1.** $\mathfrak{T}^\pi$ *is non-expansive under $MI_\Pi^\infty$ for any non-trivial joint distribution $\Pi$.*

Please refer to Appendix C for the proof, where we investigate both (**S**) and (**I**) properties. The non-trivial $\Pi$ rules out the independence case of $\mu$ and $\nu$, where $\mathrm{KL}(\Pi | \mu \otimes \nu)$ would degenerate

to zero. Although the non-expansive nature of the introduced regularization term, as shown in Proposition 1, may potentially slow the convergence in Sinkhorn divergence compared with $W_p$ without the regularization, we will demonstrate that $\mathfrak{T}^\pi$ is still contractive under the full Sinkhorn divergence in Theorem 1. Before introducing Theorem 1, we first present the sum-invariant and a new variant of scale-sensitive properties in Proposition 2, which acts as the foundation for Theorem 1.

**Proposition 2.** *Considering $\mathcal{W}_{c,\varepsilon}$ with the unrectified kernel $k_\alpha := -\|x-y\|^\alpha$ as $-c$ ($\alpha > 0$) and a scaling factor $a \in (0,1)$, $\mathcal{W}_{c,\varepsilon}$ is sum-invariant (**I**) and satisfies $\mathcal{W}_{c,\varepsilon}(a\mu, a\nu) \leq \Delta_\varepsilon(a,\alpha)\mathcal{W}_{c,\varepsilon}(\mu,\nu)$ (**S**) with a scaling constant $\Delta_\varepsilon(a,\alpha) \in (|a|^\alpha, 1)$ for any $\mu$ and $\nu$ in a finite set of probability measures.*

*Proof Sketch.* The detailed proof is provided in Appendix D. Let $\Pi^*$ be the optimal coupling of $\mathcal{W}_{c,\varepsilon}$, we define a ratio $\lambda_\varepsilon(\mu,\nu)$ that satisfies $\lambda_\varepsilon(\mu,\nu) = \frac{\varepsilon \mathrm{KL}(\Pi^*|\mu \otimes \nu)}{\mathcal{W}_{c,\varepsilon}} \in (0,1)$ for a generally non-zero $\mathcal{W}_{c,\varepsilon}$. The ratio $\lambda_\varepsilon(\mu,\nu)$ measures the proportion of the entropic regularization term over the whole loss term $\mathcal{W}_{c,\varepsilon}$. Therefore, the contraction factor $\Delta_\varepsilon(a,\alpha)$ is defined as $\Delta_\varepsilon(a,\alpha) = |a|^\alpha(1 - \sup_{\mu,\nu}\lambda_\varepsilon(\mu,\nu)) + \sup_{U,V}\lambda_\varepsilon(\mu,\nu)) \in (|a|^\alpha, 1)$ with $\sup_{\mu,\nu}\lambda_\varepsilon(\mu,\nu) < 1$, which is determined by the scale factor $a$, the order $\alpha$, the hyperparameter $\varepsilon$, and the set of interested probability measures.

**Contraction Guarantee and Interpolation Relationship.** Proposition 2 reveals that $\mathcal{W}_{c,\varepsilon}$ with an unrectified kernel satisfies (**I**) and a variant of (**S**) properties. While the scaling constant $\Delta_\varepsilon(a,\alpha)$ in (**S**) has a complicated form, it remains strictly less than one, even considering a non-expansive nature of the entropic regularization as shown in Proposition 1. We denote the supremal form of Sinkhorn divergence as $\overline{\mathcal{W}}_{c,\varepsilon}^\infty(\mu,\nu) : \overline{\mathcal{W}}_{c,\varepsilon}^\infty(\mu,\nu) = \sup_{(s,a)\in\mathcal{S}\times\mathcal{A}} \mathcal{W}_{c,\varepsilon}(\mu(s,a),\nu(s,a))$. In Theorem 1, we will integrate all these properties to demonstrate the contraction property of distributional dynamic programming under $\overline{\mathcal{W}}_{c,\varepsilon}$, specifically highlighting the interpolation property of Sinkhorn divergence between MMD and Wasserstein distance in the context of distributional RL.

**Theorem 1.** *Considering $\overline{\mathcal{W}}_{c,\varepsilon}(\mu,\nu)$ with an unrectified kernel $k_\alpha := -\|x-y\|^\alpha$ as $-c$ ($\alpha > 0$), where $\mu, \nu \in$ the distribution set of $\{Z^\pi(s,a)\}$ for $s \in \mathcal{S}$, $a \in \mathcal{A}$ in a finite MDP. We define the ratio $\overline{\lambda}_\varepsilon(\mu,\nu)$ as $\overline{\lambda}_\varepsilon(\mu,\nu) = \frac{\varepsilon \mathrm{KL}(\Pi^*|\mu \otimes \nu)}{\overline{\mathcal{W}}_{c,\varepsilon}(\mu,\nu)} \in (0,1)$ with $\sup_{\mu,\nu}\overline{\lambda}_\varepsilon(\mu,\nu) < 1$. Then, we have:*

*(1) ($\varepsilon \to 0$) $\overline{\mathcal{W}}_{c,\varepsilon}(\mu,\nu) \to 2W_\alpha^\alpha(\mu,\nu)$. When $\varepsilon = 0$, $\mathfrak{T}^\pi$ is $\gamma^\alpha$-contractive under $\overline{\mathcal{W}}_{c,\varepsilon}^\infty$.*

*(2) ($\varepsilon \to +\infty$) $\overline{\mathcal{W}}_{c,\varepsilon}(\mu,\nu) \to MMD_{k_\alpha}^2(\mu,\nu)$. When $\varepsilon = +\infty$, $\mathfrak{T}^\pi$ is $\gamma^\alpha$-contractive under $\overline{\mathcal{W}}_{c,\varepsilon}^\infty$.*

*(3) ($\varepsilon \in (0,+\infty)$), $\mathfrak{T}^\pi$ is at least $\overline{\Delta}_\varepsilon(\gamma,\alpha)$-**contractive** under $\overline{\mathcal{W}}_{c,\varepsilon}^\infty$, where $\overline{\Delta}_\varepsilon(\gamma,\alpha)$ is an MDP-dependent constant defined as $\overline{\Delta}_\varepsilon(\gamma,\alpha) = \gamma^\alpha(1 - \sup_{\mu,\nu}\overline{\lambda}_\varepsilon(\mu,\nu)) + \sup_{\mu,\nu}\overline{\lambda}_\varepsilon(\mu,\nu)) \in (\gamma^\alpha, 1)$.*

*Proof Sketch.* The detailed proof of Theorem 1 can be found in Appendix E. Theorem 1 (1) and (2) are follow-up conclusions in terms of the convergence behavior of $\mathfrak{T}^\pi$ based on the interpolation relationship between Sinkhorn divergence with Wasserstein distance and MMD [25]. We also provide a rigorous analysis within the context of distributional RL for completeness. Our critical theoretical contribution is the part (3) for the general $\varepsilon \in (0,\infty)$, where we show that $\mathfrak{T}^\pi$ is at least a $\overline{\Delta}_\varepsilon(\gamma,\alpha)$-contractive operator. The contraction factor $\overline{\Delta}_\varepsilon(\gamma,\alpha) \in (\gamma^\alpha, 1)$ depends on the return distribution set $\{Z^\pi(s,a)\}$ of the considered MDP, and it is also a function of $\gamma, \varepsilon$ and $\alpha$. Due to the influence of the regularization term in Sinkhorn loss, $\overline{\Delta}_\varepsilon(\gamma,\alpha)$ is larger than $|\gamma|^\alpha$, the contraction factor for Wasserstein distance without the regularization. Thus, $\overline{\Delta}_\varepsilon(\gamma,\alpha)$ can be seen as an interpolation between $\gamma^\alpha$ and 1, with the coefficient $\sup_{\mu,\nu}\overline{\lambda}_\varepsilon(\mu,\nu) \in (0,1)$ defined in Theorem 1. The ratio $\overline{\lambda}_\varepsilon(\mu,\nu)$ measures the proportion of the KL regularization term relative to $\overline{\mathcal{W}}_{c,\varepsilon}$. As $\varepsilon \to 0$ or $+\infty$, $\sup_{\mu,\nu}\overline{\lambda}_\varepsilon(\mu,\nu) \to 0$, leading to $\gamma^\alpha$-contraction. This aligns with parts (1) and (2).

**Consistency with Existing Contraction Conclusions.** As Sinkhorn divergence interpolates between Wasserstein distance and MMD, its contraction property for $\varepsilon \in [0,\infty]$ also aligns well with the existing distributional RL algorithms when $c = -k_\alpha$. It is worth noting that using Gaussian kernels in the cost function does not yield concise or consistent contraction results like those in Theorem 1 (3). This conclusion is consistent with MMD-DQN [37] ($\varepsilon \to +\infty$), where $\mathfrak{T}^\pi$ is generally not a contraction operator under MMD with Gaussian kernels, as counterexamples exist (Theorem 2) in [37]. Guided by our theoretical results, we employ the rectified kernel $k_\alpha$ as the cost function and set $\alpha = 2$ in our experiments, ensuring that $\mathfrak{T}^\pi$ retains the contraction property guaranteed by Theorem 1 (3). In Table 1, we also summarize the main properties of distribution divergences in typical distributional RL algorithms, including the convergence rate of $\mathfrak{T}^\pi$ and sample complexity,

| Algorithm | $d_p$ Distribution Divergence | Representation $Z_\theta$ | Convergence Rate of $\mathfrak{T}^\pi$ | Sample Complexity of $d_p$ |
|---|---|---|---|---|
| C51 | Cramér distance | Categorical Distribution | $\sqrt{\gamma}$ | |
| QR-DQN-1 | Wasserstein distance | Quantiles | $\gamma$ | $\mathcal{O}(n^{-\frac{1}{d}})$ |
| MMD-DQN | MMD | Samples | $\gamma^{\alpha/2}$ $(k_\alpha)$ | $\mathcal{O}(n^{-1})$ |
| SinkhornDRL (ours) | Sinkhorn divergence $(c = -k_\alpha)$ | Samples | $\gamma$ $(\varepsilon \to 0)$ $\gamma^{\alpha/2}$ $(\varepsilon \to \infty)$ | $\mathcal{O}(n^{\frac{\varepsilon}{\varepsilon\lfloor d/2\rfloor \sqrt{n}}})$ $(\varepsilon \to 0)$ $\mathcal{O}(n^{-\frac{1}{2}})$ $(\varepsilon \to \infty)$ |

Table 1: Properties of different distribution divergences in typical distributional RL algorithms. $d$ is the sample dimension and $\kappa = 2\beta d + \|c\|_\infty$, where the cost function $c$ is $\beta$-Lipschitz [24]. Sample complexity is improved to $\mathcal{O}(1/n)$ using the kernel herding technique [10] in MMD.

i.e., the convergence rate of a given metric between a measure and its empirical counterpart as a function of the number of samples $n$.

## 4.2 Extension to Multi-dimensional Return Distributions

As the ability to extend to the multi-dimensional reward setting is one of the major advantages of SinkhornDRL over quantile regression-based algorithms, we next demonstrate that the joint distributional Bellman operator in the multi-dimensional reward case is contractive under Sinkhorn divergence $\overline{\mathcal{W}}_{c,\varepsilon}^\infty$. First, we define a $d$-dimensional reward function as $\mathbf{R}: \mathcal{S} \times \mathcal{A} \to P(\mathbb{R}^d)$, where $d$ represents the number of reward sources. Consequently, we have joint return distributions of the $d$-dimensional return vector $\mathbf{Z}^\pi(s, a) = \sum_{t=0}^\infty \gamma^t \mathbf{R}(s_t, a_t)$, where $\mathbf{Z}^\pi(s, a) = (Z_1^\pi(s, a), \cdots, Z_d^\pi(s, a))^\top$. The joint distributional Bellman operator $\mathfrak{T}_d^\pi$ applied on the joint distribution of the random vector $\mathbf{Z}(s, a)$ is defined as $\mathfrak{T}_d^\pi \mathbf{Z}(s, a) \overset{D}{:=} \mathbf{R}(s, a) + \gamma \mathbf{Z}(s', a')$, where $s' \sim P(\cdot|s, a)$, $a' \sim \pi(\cdot|s')$.

**Corollary 1.** *For two joint distributions $\mathbf{Z}_1$ and $\mathbf{Z}_2$, $\mathfrak{T}_d^\pi$ is $\overline{\Delta}_\varepsilon(\gamma, \alpha)$-contractive under $\overline{\mathcal{W}}_{c,\varepsilon}^\infty$, i.e.,*

$$\overline{\mathcal{W}}_{c,\varepsilon}^\infty(\mathfrak{T}^\pi \mathbf{Z}_1, \mathfrak{T}^\pi \mathbf{Z}_2) \leq \overline{\Delta}_\varepsilon(\gamma, \alpha) \overline{\mathcal{W}}_{c,\varepsilon}^\infty(\mathbf{Z}_1, \mathbf{Z}_2). \tag{4}$$

Please refer to Appendix F for the proof. The contraction guarantee of Sinkhorn divergence enables us to effectively deploy our SinkhornDRL algorithm in various RL tasks that involve multiple sources of rewards [32, 15], hybrid reward architecture [50, 30], or sub-reward structures after reward decomposition [31, 57]. We compare SinkhornDRL with MMD-DQN in multiple reward sources setting in Section 5.3, where SinkhornDRL significantly outperforms MMD-DQN by leveraging its ability to capture richer data geometry, a key advantage of optimal transport distances.

## 4.3 SinkhornDRL Algorithm and Approximation

**Equipping Sinkhorn Divergence and Particle Representation.** The key to applying Sinkhorn divergence in distributional RL is to leverage the Sinkhorn loss $\overline{\mathcal{W}}_{c,\varepsilon}$ to measure the distance between the current action-return distribution $Z_\theta(s, a)$ and the target distribution $\mathfrak{T}^\pi Z_\theta(s, a)$. This yields $\overline{\mathcal{W}}_{c,\varepsilon}(Z_\theta(s, a), \mathfrak{T}^\pi Z_\theta(s, a))$ for each $s, a$ pair. For the representation of $Z_\theta(s, a)$, we employ the unrestricted statistics, i.e., deterministic samples, akin to MMD-DQN, instead of predefined statistic functionals like quantiles in QR-DQN or categorical distributions in C51. More concretely, we use neural networks to generate samples to approximate the return distributions, expressed as $Z_\theta(s, a) := \{Z_\theta(s, a)_i\}_{i=1}^N$, where $N$ is the number of generated samples. We refer to these samples $\{Z_\theta(s, a)_i\}_{i=1}^N$ as *particles*. We then use the Dirac mixture $\frac{1}{N} \sum_{i=1}^N \delta_{Z_\theta(s, a)_i}$ to approximate the

---

**Algorithm 1** Generic Sinkhorn distributional RL Update

**Require**: Number of generated samples $N$, the cost function $c$, hyperparameter $\varepsilon$ and the target network $Z_{\theta^*}$.

**Input**: Sample transition $(s, a, r', s')$

1: **Policy evaluation**: $a^* \sim \pi(\cdot|s')$ or **Control**: $a^* \leftarrow \arg\max_{a' \in \mathcal{A}} \frac{1}{N} \sum_{i=1}^N Z_\theta(s', a')_i$
2: $\mathfrak{T} Z_i \leftarrow r + \gamma Z_{\theta^*}(s', a^*)_i, \forall 1 \leq i \leq N$

**Output**: $\overline{\mathcal{W}}_{c,\varepsilon}\left(\{Z_\theta(s, a)_i\}_{i=1}^N, \{\mathfrak{T} Z_j\}_{j=1}^N\right)$

---

true density function of $Z^\pi(s, a)$, thus minimizing the Sinkhorn divergence between the approximate distribution and its distributional Bellman target. A generic Sinkhorn distributional RL algorithm with particle representation is provided in Algorithm 1.

**Efficient Approximation via Sinkhorn Iterations with Guarantee.** By introducing an entropy regularization, Sinkhorn divergence renders optimal transport computationally feasible, especially in the high-dimensional space, via efficient algorithms, e.g., Sinkhorn Iterations [45, 25]. Notably, Sinkhorn iteration with $L$ steps yields a differentiable and solvable efficient loss function as the main burden is the matrix-vector multiplication, which streams well on the GPU by simply adding extra differentiable layers on the typical deep neural network, such as a DQN architecture. *It has been proven that Sinkhorn iterations asymptotically converge to the true loss in a linear rate* [25, 22, 12, 27]. We provide a detailed description of Sinkhorn iterations in Algorithm 2 and a full version in Algorithm 3 of Appendix G. In practice, selecting proper values of $L$ and $\varepsilon$ is crucial. To this end, we conduct a rigorous sensitivity analysis, detailed in Section 5.

**Remark: Relationship with IQN and FQF.** In the realm of distributional RL algorithms, it is important to highlight that QR-DQN and MMD-DQN are direct counterparts to SinkhornDRL within the first dimension of algorithmic evolution. In contrast, IQN and FQF enhance QR-DQN and position them in the second modeling dimension, which are orthogonal to our work. As discussed in [37], the techniques from IQN and FQF can naturally extend both MMD-DQN and SinkhornDRL. For instance, we can implicitly generate $\{Z_\theta(s, a)_i\}_{i=1}^N$ by applying a neural network to $N$ samples of a base sampling distribution, as in IQN. We can also use a proposal network to learn the weights of each generated sample as in FQF. We leave these modeling extensions as future works and our current study focuses on rigorously investigating the simplest modeling choice via Sinkhorn divergence.

## 5 Experiments

We substantiate the effectiveness of SinkhornDRL as described in Algorithm 1 on the entire 55 Atari 2600 games. Without increasing the model capacity for a fair comparison, we leverage the same architecture as QR-DQN and MMD-DQN, and replace the quantiles output in QR-DQN with $N$ particles (samples). In contrast to MMD-DQN, SinkhornDRL only changes the distribution divergence from MMD to Sinkhorn divergence. As such, the potential performance improvement of our algorithm is directly attributed to the theoretical advantages of Sinkhorn divergence over MMD.

**Baseline Implementation.** We choose DQN [36] and three typical distributional RL algorithms as classic baselines, including C51 [5], QR-DQN [14] and MMD-DQN [37]. For a fair comparison, we build SinkhornDRL and all baselines based on a well-accepted PyTorch implementation[2] of distributional RL algorithms. We re-implement MMD-DQN based on its original TensorFlow implementation[3], and keep the same setting. For example, our MMD-DQN still employs Gaussian kernels $k_h(x, y) = \exp(-(x-y)^2/h)$ with the same kernel mixture trick covering a range of bandwidths $h$ as adopted in MMD-DQN [37].

**SinkhornDRL Implementation and Hyperparameter Settings.** For a fair comparison with QR-DQN, C51, and MMD-DQN, we use the same hyperparameters: the number of generated samples $N = 200$, Adam optimizer with lr $= 0.00005$, $\epsilon_{\text{Adam}} = 0.01/32$. In SinkhornDRL, we choose the number of Sinkhorn iterations $L = 10$ and smoothing hyperparameter $\varepsilon = 10.0$ in Section 5.1 after conducting sensitivity analysis in Section 5.2. Guided by the contraction guarantee analyzed in Theorem 1, we use *the unrectified kernel*, specifically setting $-c = k_\alpha$ and choosing $\alpha = 2$. This choice ensures *our implementation is consistent with the theoretical results regarding the contraction guarantee in Theorem 1 (3)*. We evaluate all algorithms on 55 Atari games, averaging results over three seeds. The shade in the learning curves of each game represents the standard deviation.

### 5.1 Performance of SinkhornDRL

**Learning Curves of Human Normalized Scores (HNS).** We compare the learning curves of the Mean, Median, and Interquartile Mean (IQM) [1] across all considered distributional RL algorithms in Figure 1 summarized over 55 Atari games. The IQM (x%) computes the mean from the $x\%$ to $(1 - x)\%$ range of HNS, providing a robust alternative to the Mean that mitigates the impact of

---

[2]https://github.com/ShangtongZhang/DeepRL
[3]https://github.com/thanhnguyentang/mmdrl

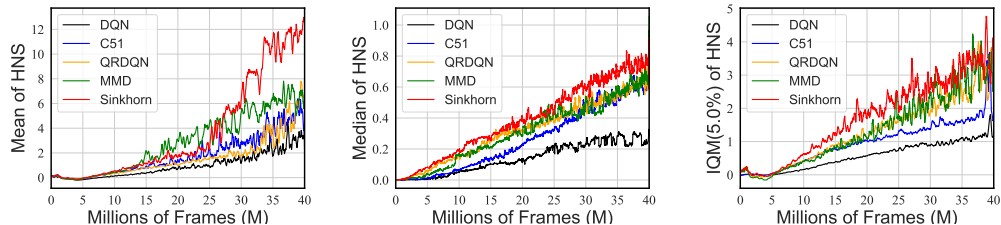

Figure 1: Mean (left), Median (middle), and IQM (5%) (right) of Human-Normalized Scores (HNS) summarized over 55 Atari games. We run 3 seeds for each algorithm.

extremely high scores on specific games and is more statistically efficient than the Median. For computational feasibility, we evaluate the algorithms over 40M training frames. Our findings reveal that SinkhornDRL achieves state-of-the-art performance in terms of mean, median, and IQM (5%) of HNS across most training phases. Notably, SinkhornDRL exhibits slower convergence during the early training phase, as indicated by the Mean of HNS (left panel of Figure 1). This slower initial convergence can be explained by the slower contraction factor $\overline{\Delta}_\varepsilon(\gamma, \alpha) > \gamma^\alpha$ in Theorem 1, as opposed to MMD-DQN. To ensure the reliability of our results, we also provide the learning curves for each Atari game in Figure 6 in Appendix I. Furthermore, a table summarizing all raw scores is available in Table 3 in Appendix J. This table highlights that SinkhornDRL achieves the highest numbers of best and second-best performance of all games among all baseline algorithms. A summary table of Mean, IQM, and Median HNS is also given in Table 2 of Appendix H. Overall, we conclude that SinkhornDRL generally outperforms existing distributional RL algorithms.

**Ratio Improvement Analysis across All Games.** Given the interpolation nature of Sinkhorn divergence between Wasserstein distance and MMD, as analyzed in Theorem 1, a pertinent question arises: *In which environments does SinkhornDRL potentially perform better?* We empirically address this question by conducting a ratio improvement comparison between SinkhornDRL and both QR-DQN and MMD-DQN across all games. Figure 2 showcases that SinkhornDRL surpasses both QR-DQN and MMD-DQN in more than half of the games and significantly excels at them in a large proportion of games. Notably, *the games where SinkhornDRL achieves considerable improvement tend to have larger action spaces and more complex dynamics*. In particular, as illustrated in Figure 2, these games include Venture, Seaquest, Solaris, Tennis, Phoenix, Atlantis, and Zaxxon. Most of these games have an 18-dimensional action space and intricate dynamics, except for Atlantis, which has a 4-dimensional action space and simpler dynamics where MMD-DQN is substantially inferior to SinkhornDRL. For a detailed comparison, we provide the features of all games, including the number of action spaces, and complexity of environment dynamics in Table 4 of Appendix K.

In summary, compared with QR-DQN, the empirical success of SinkhornDRL can be attributed to several key factors: 1. *Enhanced return distribution representation:* SinkhornDRL captures return

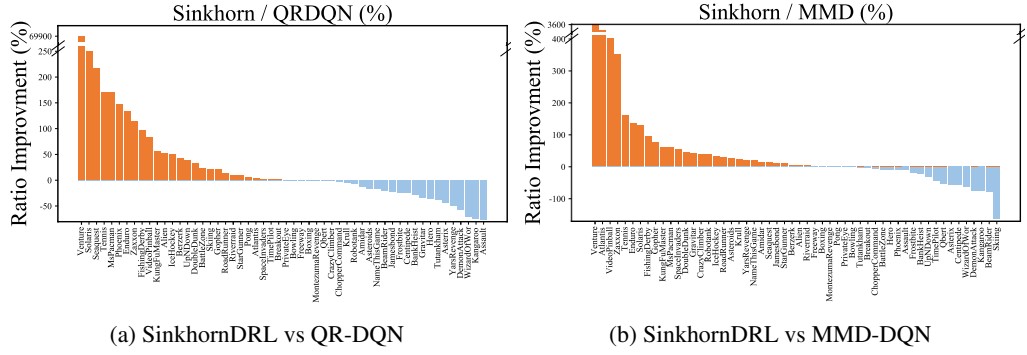

(a) SinkhornDRL vs QR-DQN    (b) SinkhornDRL vs MMD-DQN

Figure 2: Ratio improvement of return for SinkhornDRL over QR-DQN (left) and MMD-DQN (right) averaged over 3 seeds. The ratio improvement is calculated by (SinkhornDRL - QR-DQN) / QR-DQN in (a) and (SinkhornDRL - MMD-DQN) / MMD-DQN in (b), respectively.

distribution characteristics more accurately by directly using samples, avoiding the non-crossing issue of learned quantile curves or the potential limitations of quantile representation. *2. Smooth transport plan and stable convergence.* The induced smoother transport plan (see Appendix A for visualization) and the inherent smoothness of Sinkhorn divergence contribute to more stable convergence, leading to performance improvement. In contrast to MMD-DQN, the benefits of SinkhornDRL arise from its richer data representation capability when comparing return distributions, rooted in the OT nature. This is in comparison to the potentially restricted kernel-specific distances, such as MMD.

## 5.2 Sensitivity Analysis and Computational Cost

**Sensitivity Analysis.** In practice, a proper $\varepsilon$ is preferable as an overly large or small $\varepsilon$ will lead to numerical instability of Sinkhorn iterations in Algorithm 2 (see the discussion in Section 4.4 of [39]), therefore worsening its performance, as shown in Figure 3 (a). This implies that the potential interpolation nature of limiting behaviors between SinkhornDRL with QR-DQN and MMD-DQN revealed in Theorem 1 may not be able to be rigorously verified in numerical experiments. SinkhornDRL also requires a proper number of iterations $L$ and samples $N$. For example, a small $N$, e.g., $N = 2$ in Seaquest in Figure 3 (b) leads to the divergence of algorithms, while an overly large $N$ can degrade the performance and meanwhile increases the computational burden (Appendix L.2). We conjecture that using larger networks to represent more samples is more likely to suffer from overfitting, yielding the instability in the RL training [7]. Therefore, we choose $N = 200$ to attain favorable performance and guarantee computational effectiveness simultaneously. We provide a more detailed sensitivity analysis and more results on StarGunner and Zaxxon in Appendix L.1.

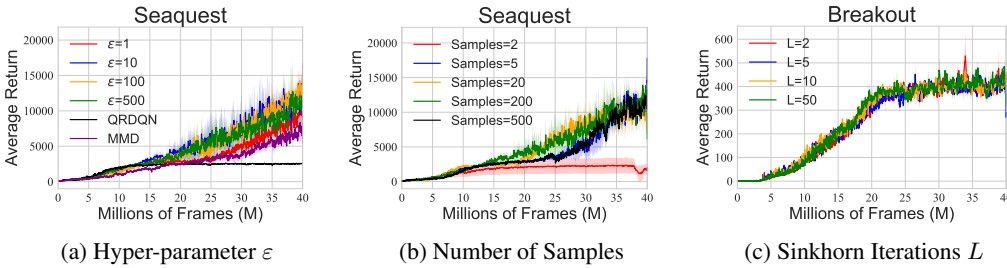

(a) Hyper-parameter $\varepsilon$        (b) Number of Samples        (c) Sinkhorn Iterations $L$

Figure 3: Sensitivity analysis of SinkhornDRL on Breakout and Seaquest in terms of $\varepsilon$, number of samples, and number of iteration $L$. Learning curves are reported over three seeds.

**Computation Cost.** In terms of the computation cost, SinkhornDRL slightly increases the computational overhead compared with C51, QR-DQN, and MMD-DQN. For instance, SinkhornDRL increases the average computational cost compared with MMD-DQN by around 20%. Due to the space limit, we provide more computation cost comparison in terms of $L$ and $N$ in Appendix L.2.

## 5.3 Modeling Joint Return Distribution for Multi-Dimensional Reward Functions

Many RL tasks involve modeling multivariate return distributions. Following the multi-dimensional reward setting in [57], we compare SinkhornDRL with MMD-DQN on six Atari games with multiple sources of rewards. In these tasks, the primitive scalar-based rewards are decomposed into reward vectors based on the respective reward structures (see Appendix M for more details). Figure 4 showcases that SinkhornDRL outperforms MMD-DQN in most cases for multi-dimensional reward functions. Of particular note, it remains an open question to directly approximate multi-dimensional Wasserstein distances via quantile regression or other efficient algorithms, particularly in RL tasks.

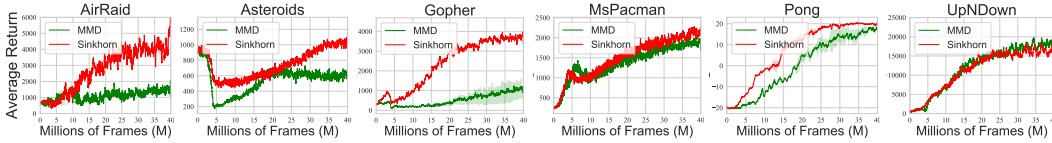

Figure 4: Performance of SinkhornDRL on six Atari games with multi-dimensional reward functions.

# 6 Conclusion, Limitations and Future Work

In this work, we propose a novel family of distributional RL algorithms based on Sinkhorn divergence that accomplishes competitive performance compared with the typical distributional RL algorithms on the Atari games suite. Theoretical results about the properties of this regularized Wasserstein loss and its convergence guarantee in the context of RL are provided with rigorous empirical verification.

**Limitations.** While SinkhornDRL achieves competitive performance, it relatively increases the computational cost and requires tuning additional hyperparameters. This hints that the enhanced performance offered by SinkhornDRL may come with slightly greater efforts in practical deployment. Additionally, it remains elusive for a deeper connection between the theoretical properties of divergences and the practical performance of distributional RL algorithms given a specific environment.

**Future work.** Along the two dimensions of distributional RL algorithm evolution, we can further improve Sinkhorn distributional RL by incorporating implicit generative models, including parameterizing the cost function and increasing model capacity. Moreover, Sinkhorn distributional RL also opens a door for new applications of Sinkhorn divergence and more optimal transport approaches in RL. It also becomes increasingly crucial to design a quantitative criterion for a given environment to recommend the choice of a specific distribution divergence before conducting costly experiments.

## Acknowledgements

Yingnan Zhao and Ke Sun were supported by the State Scholarship Fund from China Scholarship Council (No:202006120405 and No:202006010082). Bei Jiang and Linglong Kong were partially supported by grants from the Canada CIFAR AI Chairs program, the Alberta Machine Intelligence Institute (AMII), and Natural Sciences and Engineering Council of Canada (NSERC), and Linglong Kong was also partially supported by grants from the Canada Research Chair program from NSERC. The authors express their gratitude for the insightful feedback provided by all reviewers and the area chairs, which significantly enhanced the initial version of this paper.

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

# Appendix

## Table of Contents

# A  Smoother Transport Plan via Sinkhorn Divergence by Increasing $\varepsilon$

We visualize the optimal transport plans by solving Sinkhorn divergence with different $\varepsilon$ in well-trained SinkhornDRL models across three games in Figure 5 We evaluate (randomly selected 64) current and target state features to be compared and then apply t-SNE to reduce their dimension from 512 to 2 associated with a normalization for visualization. In each game of Figure 5, as we increase the regularization strength $\varepsilon$ (from right to left), the resulting transport plans tend to be smoother, less concentrated, and more uniformly distributed by transporting the point mass between two distributions (in red and blue).

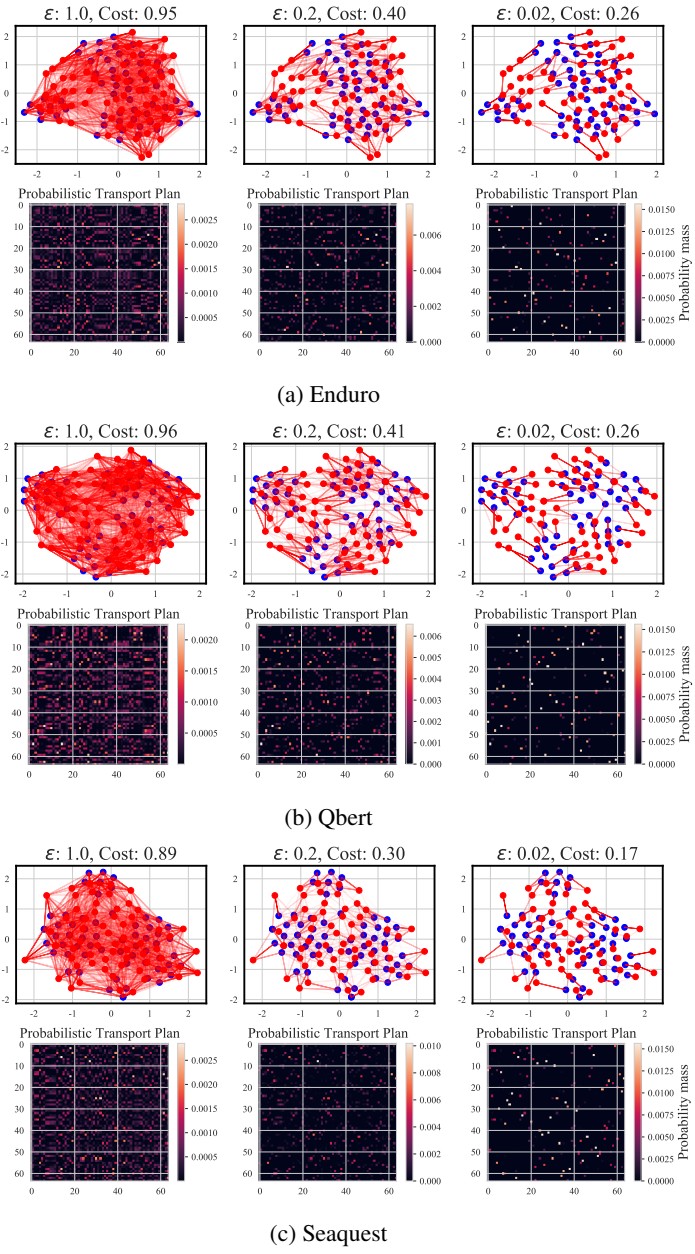

(a) Enduro

(b) Qbert

(c) Seaquest

Figure 5: Optimal transport plans for via Sinkhorn Iterations in SinkhornDRL on three Atari games. The first row denotes the (two-dimensional) spatial transport plans across different data points, while the second row represents the heat map of the obtained transport plan (optimal coupling).

# B  Definition of Distribution Divergences and Contraction Properties

**Definition of distances.** Given two random variables $X$ and $Y$, one-dimensional $p$-Wasserstein metric $W_p$ between the distributions of $X$ and $Y$ has a simplified form via the quantile functions:

$$W_p(X, Y) = \left( \int_0^1 \left| F_X^{-1}(\omega) - F_Y^{-1}(\omega) \right|^p d\omega \right)^{1/p} = \| F_X^{-1} - F_Y^{-1} \|_p, \tag{5}$$

which $F^{-1}$ is the quantile function, also known as inverse cumulative distribution function, of a random variable with the cumulative distribution function as $F$. The supremal form of $W_p$, denoted by $W_p^\infty$, is defined as

$$W_p^\infty(\mu, \nu) = \sup_{(s,a) \in \mathcal{S} \times \mathcal{A}} W_p^\infty(\mu(s,a), \nu(s,a)). \tag{6}$$

Further, $\ell_p$ distance [19] is defined as

$$\ell_p(X, Y) := \left( \int_{-\infty}^\infty |F_X(\omega) - F_Y(\omega)|^p \, d\omega \right)^{1/p} = \| F_X - F_Y \|_p. \tag{7}$$

The $\ell_p$ distance and Wasserstein metric are identical at $p = 1$, but are otherwise distinct. Note that when $p = 2$, $\ell_p$ distance is also called Cramér distance [6] $d_C(X, Y)$. Also, Cramér distance has a different representation given by

$$d_C(X, Y) = \mathbb{E}|X - Y| - \frac{1}{2}\mathbb{E}|X - X'| - \frac{1}{2}\mathbb{E}|Y - Y'|, \tag{8}$$

where $X'$ and $Y'$ are the i.i.d. copies of $X$ and $Y$. Energy distance [49, 59] is a natural extension of Cramér distance to the multivariate case, which is defined as

$$d_E(\mathbf{X}, \mathbf{Y}) = \mathbb{E}\|\mathbf{X} - \mathbf{Y}\| - \frac{1}{2}\mathbb{E}\|\mathbf{X} - \mathbf{X}'\| - \frac{1}{2}\mathbb{E}\|\mathbf{Y} - \mathbf{Y}'\|, \tag{9}$$

where $\mathbf{X}$ and $\mathbf{Y}$ are multivariate. Moreover, the energy distance is a special case of the maximum mean discrepancy (MMD), which is formulated as

$$\text{MMD}(\mathbf{X}, \mathbf{Y}; k) = \left( \mathbb{E}\left[ k\left(\mathbf{X}, \mathbf{X}'\right) \right] + \mathbb{E}\left[ k\left(\mathbf{Y}, \mathbf{Y}'\right) \right] - 2\mathbb{E}[k(\mathbf{X}, \mathbf{Y})] \right)^{1/2}, \tag{10}$$

where $k(\cdot, \cdot)$ is a continuous kernel on $\mathcal{X}$. In particular, if $k$ is a trivial kernel, also called the unrectified kernel, MMD degenerates to energy distance. Additionally, we further define the supreme MMD, which is a functional $\mathcal{P}(\mathcal{X})^{\mathcal{S} \times \mathcal{A}} \times \mathcal{P}(\mathcal{X})^{\mathcal{S} \times \mathcal{A}} \to \mathbb{R}$ formulated as

$$\text{MMD}_\infty(\mu, \nu) = \sup_{(s,a) \in \mathcal{S} \times \mathcal{A}} \text{MMD}_\infty(\mu(s,a), \nu(s,a)). \tag{11}$$

We further summarize the convergence rates of the distributional Bellman operator $\mathfrak{T}^\pi$ under different distribution divergences.

- $\mathfrak{T}^\pi$ is $\gamma$-contractive under the supreme form of Wassertein distance $W_p$.

- $\mathfrak{T}^\pi$ is $\gamma^{1/p}$-contractive under the supreme form of $\ell_p$ distance.

- $\mathfrak{T}^\pi$ is $\gamma^{\alpha/2}$-contractive under $\text{MMD}_\infty$ with the kernel $k_\alpha(x, y) = -\|x - y\|^\alpha, \forall \alpha > 0$.

**Proof of Contraction in Distributional Dynamic Programming.**

- Contraction under the supreme form of Wasserstein distance is provided in Lemma 3 [5].

- Contraction under supreme form of $\ell_p$ distance can refer to Theorem 3.4 [19].

- Contraction under $\text{MMD}_\infty$ is provided in Lemma 6 [37].

## C   Proof of Proposition 1

*Proof.* We denote two marginal random variables $U$ and $V$ with the pdf $\mu(x)$ and $\nu(y)$. We next denote the $p_\Pi(x, y)$ as the pdf for $\Pi$ in $\text{MI}_\Pi(U, V) = \text{KL}(\Pi|U \otimes V)$. We first prove that the $\text{MI}_\Pi(U, V)$ is sum-invariant, which is based on the dual form of KL divergence via the variational representation [17, 2]:

$$D_{KL}(X, Y) = \sup_{f \in \mathcal{L}^b} \{\mathbb{E}_X[f(x)] - \log\left(\mathbb{E}_Y\left[e^{f(y)}\right]\right)\}, \tag{12}$$

where $\mathcal{L}^b$ is the space of bounded measurable functions. The mutual information involves two-dimensional random variables in the KL divergence. Let $U' = a + U$ and $V = a + V$ with pdf $\mu'$ and $\nu'$, we denote the joint distribution with margins $\mu'(x) = \mu(x - a)$ and $\nu'(y) = \nu(y - a)$ as $\Pi'(x, y)$ whose pdf $p_{\Pi'}$ satisfies $p_{\Pi'}(x, y) = p_\Pi(x - a, y - a)$. Based on the two-dimensional variational representation of KL divergence $\text{MI}_\Pi(U, V) = \sup_{f \in \mathcal{L}^b} \{\mathbb{E}_\Pi[f(x, y)] - \log\left(\mathbb{E}_{U,V}\left[e^{f(x,y)}\right]\right)\}$, we have:

$$\text{MI}_\Pi(A + U, A + V)$$
$$= \sup_{f \in \mathcal{L}^b} \{\mathbb{E}_{\Pi'}[f(x, y)] - \log\left(\mathbb{E}_{A+U,A+V}\left[e^{f(x,y)}\right]\right)\}$$
$$\stackrel{(a)}{=} \sup_{f \in \mathcal{L}^b} \{\mathbb{E}_A\left[\mathbb{E}_{\Pi(x-a,y-a)}[f(x, y)]\right] - \log\left(\mathbb{E}_A\left[\mathbb{E}_{a+U,a+V}\left[e^{f(x,y)}\right]\right]\right)\}$$
$$= \sup_{f \in \mathcal{L}^b} \{\mathbb{E}_A\left[\mathbb{E}_{\Pi(x,y)}[f(x+a, y+a)]\right] - \log\left(\mathbb{E}_A\left[\mathbb{E}_{U,V}\left[e^{f(x+a,y+a)}\right]\right]\right)\}$$
$$\stackrel{(b)}{\leq} \sup_{f \in \mathcal{L}^b} \{\mathbb{E}_A\mathbb{E}_\Pi[f(x+a, y+a)] - \mathbb{E}_A \log\left(\mathbb{E}_{U,V}\left[e^{f(x+a,y+a)}\right]\right)\} \tag{13}$$
$$= \sup_{f \in \mathcal{L}^b} \{\mathbb{E}_A[\mathbb{E}_\Pi[f(x+a, y+a)] - \log\left(\mathbb{E}_{U,V}\left[e^{f(x+a,y+a)}\right]\right)]\}$$
$$\stackrel{(c)}{\leq} \mathbb{E}_A \sup_{f \in \mathcal{L}^b} \{\mathbb{E}_\Pi[f(x+a, y+a)] - \log\left(\mathbb{E}_{U,V}\left[e^{f(x+a,y+a)}\right]\right)\}$$
$$\stackrel{(d)}{=} \mathbb{E}_A \sup_{g \in \mathcal{L}^b} \{\mathbb{E}_\Pi[g(x, y)] - \log\left(\mathbb{E}_{U,V}\left[e^{g(x,y)}\right]\right)\}$$
$$= \text{MI}_\Pi(U, V),$$

where (a) is by the independence of $A$ between $X, Y$, and the joint cdf $\Pi$. For instance, in the one-dimensional setting, we have $\mathbb{E}_{Z=A+X}[f(z)] = \int_a \int_x f(x + a) p_A(a) p_X(x) dx da = \mathbb{E}_A\left[\mathbb{E}_X[f(x + a)]\right]$. (b) and (c) are by Jensen's inequality in terms of the convex function $-\log(x)$ and $\sup_f$, and (d) is because the translated cdf is still within $\mathcal{L}^b$.

Next, we show that $\text{MI}_\Pi$ is NOT scale-sensitive or with the zero-order $\tau$. This result is directly based on the similar property of KL divergence. With a slight abuse of notations, we denote $U' = aU$ and $V' = aV$, whose pdfs are $\mu'(x) = \frac{1}{a}\mu(\frac{x}{a})$ and $\nu'(y) = \frac{1}{a}\nu(\frac{y}{a})$, respectively. The scaled joint distribution $\Pi'$ with the pdf $p_{\Pi'}$ satisfying $p_{\Pi'}(x, y) = \frac{1}{a^2}p_\Pi(x/a, y/a)$. Therefore, its marginal distributions are $\int_y \frac{1}{a^2}p_\Pi(x/a, y/a)dy = \frac{1}{a}\mu(\frac{x}{a})$ and $\int_x \frac{1}{a^2}p_\Pi(x/a, y/a)dy = \frac{1}{a}\nu(\frac{y}{a})$. We thus have the following result:

$$\text{MI}_\Pi(aU, aV) = \text{KL}(\Pi'(x, y)|U' \otimes V')$$
$$= \int p_{\Pi'}(x, y) \log \frac{p_{\Pi'}(x, y)}{\mu'(x)\nu'(y)} dx dy$$
$$= \int \frac{1}{a^2}p_\Pi(x/a, y/a) \log \frac{\frac{1}{a^2}p_\Pi(x/a, y/a)}{\frac{1}{a}\mu(x/a)\nu(y/a)} dx dy \tag{14}$$
$$= \int p_\Pi(x, y) \log \frac{p_\Pi(x, y)}{\mu(x)\nu(y)} dx dy$$
$$= \text{MI}_\Pi(U, V).$$

Putting the two properties together and given two return distributions $Z_1(s, a)$ and $Z_2(s, a)$, we have the non-expansive contraction property of the supremal form of $\text{MI}_\Pi$ as follows.

$$
\begin{aligned}
\text{MI}_\Pi^\infty(\mathfrak{T}^\pi Z_1, \mathfrak{T}^\pi Z_2) &= \sup_{s,a} \text{MI}_\Pi(\mathfrak{T}^\pi Z_1(s, a), \mathfrak{T}^\pi Z_2(s, a)) \\
&= \sup_{s,a} \text{MI}_\Pi(R(s, a) + \gamma Z_1(s', a'), R(s, a) + \gamma Z_2(s', a')) \\
&\overset{(a)}{\leq} \text{MI}_\Pi(\gamma Z_1(s', a'), \gamma Z_2(s', a')) \\
&\overset{(b)}{=} \text{MI}_\Pi(Z_1(s', a'), Z_2(s', a')) \\
&\leq \sup_{s,a} \text{MI}_\Pi(Z_1(s', a'), Z_2(s', a')) \\
&= \text{MI}_\Pi^\infty(Z_1, Z_2),
\end{aligned}
\tag{15}
$$

where (a) relies on the sum invariant property of $\text{MI}_\Pi$ and (b) utilizes the non-scale sensitive property of $\text{MI}_\Pi$. By applying the well-known Banach fixed point theorem, we have a unique return distribution when convergence of distributional dynamic programming under $\text{MI}_\Pi$ for any non-trivial joint distribution $\Pi$. $\qquad\square$

## D  Proof of Proposition 2

### D.1  Sum Invariant Property

Given two random variables $U$ and $V$ with the marginal distributions as $\mu$ and $\nu$, and a random variable $A$ that is independent of them, we aim at proving

$$
\mathcal{W}_{c,\varepsilon}(A + U, A + V) \leq \mathcal{W}_{c,\varepsilon}(U, V).
\tag{16}
$$

According to [39], we have the dual form of $\mathcal{W}_{c,\varepsilon}$:

$$
\begin{aligned}
\mathcal{W}_{c,\varepsilon}(U, V) &= \sup_{\varphi,\psi} \left\{ \int_x \varphi(x)\mu_x dx + \int_y \psi(y)\nu_y dy - \varepsilon \int_{x,y} \exp\frac{\varphi(x) + \psi(y) - c(x,y)}{\varepsilon} \mu_x \nu_y dx dy \right\} \\
&= \sup_{\varphi,\psi} \left\{ \mathbb{E}_\mu\left[\varphi(x)\right] + \mathbb{E}_\nu\left[\psi(y)\right] - \varepsilon\mathbb{E}_{\mu,\nu}\left[\exp\frac{\varphi(x) + \psi(y) - c(x,y)}{\varepsilon}\right] \right\}
\end{aligned}
\tag{17}
$$

Therefore, we have:

$$
\begin{aligned}
&\mathcal{W}_{c,\varepsilon}(A + U, A + V) \\
&= \sup_{\varphi,\psi} \left\{ \mathbb{E}_{A+U}\left[\varphi(x)\right] + \mathbb{E}_{A+V}\left[\psi(y)\right] - \varepsilon\mathbb{E}_{A+U,A+V}\left[\exp\frac{\varphi(x) + \psi(y) - c(x,y)}{\varepsilon}\right] \right\} \\
&\overset{(a)}{=} \sup_{\varphi,\psi} \left\{ \mathbb{E}_A\left[\mathbb{E}_\mu\left[\varphi(x + a)\right] + \mathbb{E}_\nu\left[\psi(y + a)\right] - \varepsilon\mathbb{E}_{\mu,\nu}\left[\exp\frac{\varphi(x + a) + \psi(y + a) - c(x,y)}{\varepsilon}\right]\right] \right\} \\
&\overset{(b)}{\leq} \mathbb{E}_A\left[\sup_{\varphi,\psi} \left\{ \mathbb{E}_\mu\left[\varphi(x + a)\right] + \mathbb{E}_\nu\left[\psi(y + a)\right] - \varepsilon\mathbb{E}_{\mu,\nu}\left[\exp\frac{\varphi(x + a) + \psi(y + a) - c(x,y)}{\varepsilon}\right]\right\} \right] \\
&\overset{(c)}{=} \sup_{f,g} \left\{ \mathbb{E}_\mu\left[f(x)\right] + \mathbb{E}_\nu\left[g(y)\right] - \varepsilon\mathbb{E}_{\mu,\nu}\left[\exp\frac{f(x) + g(y) - c(x,y)}{\varepsilon}\right] \right\} \\
&= \mathcal{W}_{c,\varepsilon}(U, V),
\end{aligned}
\tag{18}
$$

where $(a)$ relies on the same techniques used in the proof of Eq. 13 in Appendix C, (b) utilizes the Jensen inequality of $\sup$, and $(c)$ is based on the fact that the translation operator is still within the same functional space of $\varphi, \psi$.

### D.2  A Variant of Scale Sensitive Property when $c = -k_\alpha$

**General Conclusion.** Let $\Pi^*$ be the optimal coupling for $\mathcal{W}_{c,\varepsilon}$, we define a ratio $\lambda_\varepsilon(U, V) = \frac{\varepsilon\text{KL}(\Pi^*|\mu\otimes\nu)}{\mathcal{W}_{c,\varepsilon}} \in (0, 1)$ for any considered $U, V$ with measures $\mu, \nu$ to compare, where the denominator

$\mathcal{W}_{c,\varepsilon}$ is generally non-zero. We thus have the following result:

$$\mathcal{W}_{c,\varepsilon}(aU, aV) \leq \Delta_\varepsilon(a, \alpha)\mathcal{W}_{c,\varepsilon}(U, V), \tag{19}$$

where the scaling factor $\Delta_\varepsilon(a, \alpha) = |a|^\alpha(1 - \sup_{U,V} \lambda_\varepsilon(U, V)) + \sup_{U,V} \lambda_\varepsilon(U, V) \in (|a|^\alpha, 1)$ with $\sup_{U,V} \lambda_\varepsilon(U, V) > 0$. *The ratio $\lambda_\varepsilon(U, V)$ measures the proportion of the entropic regularization term over the whole divergence term $\mathcal{W}_{c,\varepsilon}$*, i.e., $\lambda_\varepsilon(U, V) = \frac{\varepsilon \mathrm{KL}(\Pi^*|\mu \otimes \nu)}{\mathcal{W}_{c,\varepsilon}} \in (0, 1)$. Under the mild assumption of a finite set of probability measures, we have $\sup_{U,V} \lambda_\varepsilon(U, V) > 0$. To elaborate the reason behind it, we first know that $\lambda_\varepsilon(U, V) < 1$ for any $U$ and $V$ with their measures on the probability measure set. If this set is finite, the ratio set that contains all $\{\lambda_\varepsilon(U, V)\}$ is also finite. Based on the fact that the real set is dense, we can directly find a positive lower bound $\lambda^*$ for the ratio set, such that $\{\lambda_\varepsilon(U, V)\} \leq \lambda^* < 1$. This implies that $\sup_{U,V} \lambda_\varepsilon(U, V) = \max_{U,V} \lambda_\varepsilon(U, V) < 1$. Notably, this finite set property of the ratio avoids the extreme case that may lead to a conservative conclusion about a non-expansive distribution Bellman operator, which we will give more details later.

**Scale-sensitive Property.** By definition of Sinkhorn divergence [18, 39], the pdf of Gibbs kernel in the equivalent form of Sinkhorn divergence is $\mathcal{K}(U, V)$, which satisfies $\mathcal{K}(U, V) \propto e^{\frac{-c(x,y)}{\varepsilon}}\mu(x)\nu(y)$. In particular, the pdf of Gibbs kernel is defined as $\frac{d\mathcal{K}}{d(\mu \otimes \nu)}(x, y) = \frac{\exp(-c/\varepsilon)}{\int \exp(-c/\varepsilon)d(\mu \otimes \nu)}$, where the denominator is the normalization factor. After a scaling transformation, the pdf of $aU$ and $aV$ with respect to $x$ and $y$ would be $\frac{1}{a}\mu(\frac{x}{a})$ and $\frac{1}{a}\nu(\frac{y}{a})$. Thus $\mathcal{K}(aU, aV) \propto e^{\frac{-c(x,y)}{\varepsilon}}\frac{1}{a}\mu(\frac{x}{a})\frac{1}{a}\nu(\frac{y}{a})$. In the following proof, we use the change variable formula (multivariate version) constantly, while changing the joint pdf $\pi(x, y)$ and keep the cost function term $c(x, y)$. In particular, we denote $\Pi^*$ and $\Pi^0$ as the optimal joint distribution of $\mathcal{W}_{c,\varepsilon}(\mu, \nu)$ and $\mathcal{W}_{c,\varepsilon}(a\mu, a\nu)$. Then we have:

$$
\begin{aligned}
\mathcal{W}_{c,\varepsilon}(aU, aV) &= \int c(x, y)\mathrm{d}\Pi^0(x, y) + \varepsilon \mathrm{KL}(\Pi^0|a\mu \otimes a\nu) \\
&\leq \int c(x, y)\mathrm{d}\Pi^*(x, y) + \varepsilon \mathrm{KL}(\Pi^*|a\mu \otimes a\nu) \\
&\stackrel{c=-k_\alpha}{=} \int (x - y)^\alpha \frac{1}{a^2}\pi^*(\frac{x}{a}, \frac{y}{a})\mathrm{d}x\mathrm{d}y + \varepsilon \int \frac{1}{a^2}\pi^*(\frac{x}{a}, \frac{y}{a})\log \frac{\frac{1}{a^2}\pi^*(\frac{x}{a}, \frac{y}{a})}{\frac{1}{a^2}\mu(\frac{x}{a})\nu(\frac{y}{a})}\mathrm{d}x\mathrm{d}y \\
&= |a|^\alpha \int (x - y)^\alpha \pi^*(x, y)\mathrm{d}x\mathrm{d}y + \varepsilon \int \pi^*(x, y)\log \frac{\pi^*(x, y)}{\mu(x)\nu(y)}\mathrm{d}x\mathrm{d}y \\
&= |a|^\alpha \int (x - y)^\alpha \pi^*(x, y)\mathrm{d}x\mathrm{d}y + (|a|^\alpha + 1 - |a|^\alpha)\varepsilon \int \pi^*(x, y)\log \frac{\pi^*(x, y)}{\mu(x)\nu(y)}\mathrm{d}x\mathrm{d}y \\
&= |a|^\alpha \mathcal{W}_{c,\varepsilon}(U, V) + (1 - |a|^\alpha)\varepsilon \mathrm{KL}(\Pi^*|\mu \otimes \nu) \\
&= \Delta_\varepsilon^{U,V}(a, \alpha)\mathcal{W}_{c,\varepsilon}(U, V)
\end{aligned}
\tag{20}
$$

where $\Delta_\varepsilon^{U,V}(a, \alpha) = |a|^\alpha + (1 - |a|^\alpha)\lambda_\varepsilon(U, V) = |a|^\alpha(1 - \lambda_\varepsilon(U, V)) + \lambda_\varepsilon(U, V) \in (|a|^\alpha, 1)$ for $\varepsilon \in (0, +\infty)$ and $a < 1$ due to the fact that $\lambda_\varepsilon(U, V) \in (0, 1)$ for any non-trivial $\mathcal{W}_{c,\varepsilon}(U, V)$. The non-trivial $\mathcal{W}_{c,\varepsilon}(U, V)$ rules out the case when the regularization term is zero, e.g., $\epsilon = 0$ or the optimal coupling is the product of two margins. In other words, $\Delta_\varepsilon^{U,V}(a, \alpha)$ is a function less than 1, which depends on the two margins, including their independence and distribution similarity, the scale factor $a$ and the order $\alpha$.

**Ruling Out Extreme Cases in the Convergence via a Finite Set.** However, the fact that $\Delta_\varepsilon^{U,V}(a, \alpha) < 1$ can only guarantee a "conservative" non-expansive contraction rather than a desirable contraction of the distributional Bellman operator. This is because there will be extreme cases in the power of series in general, although it is very unlikely to occur given a certain MDP in practice. For example, denote the non-constant factor as $q_k$ for the k-th distributional Bellman update, where $q_k < 1$. We can construct a counterexample as $q_k = 1 - 1/(k + 2)^2$. In this case, $\Pi_{k=1}^{+\infty} q_k = (\frac{2}{3}\frac{4}{3})(\frac{3}{4}\frac{5}{4})\cdots > 0$ instead of the convergence to 0 and the non-zero limit can not guarantee the contraction. It also intuitively implies that iteratively applying distribution Bellman operator under $\mathcal{W}_{c,\varepsilon}$ may not lead to convergence *in general by considering all possible return distributions* given the non-constant factor $\Delta_\varepsilon^{U,V}(a, \alpha)$. Although we know these extreme cases are very unlikely to happen, we have to rule out these extreme cases for a rigorous proof. As we have the assumption

of a finite set of probability measures, the set of $\{\lambda_\varepsilon(U,V)\}$ is also finite. As the real set is dense, we can always find a positive constant that can be used as the contraction factor. Alternatively, we can directly use the $\sup_{U,V} \lambda_\varepsilon(U,V)$ as the uniform upper bound across the whole set of interested probability measures. Under this condition, we can immediately find a universal upper bound of $\Delta_\varepsilon^{U,V}(a,\alpha)$:

$$
\begin{aligned}
\sup_{U,V} \Delta_\varepsilon^{U,V}(a,\alpha) &= |a|^\alpha + (1 - |a|^\alpha)\sup_{U,V} \lambda_\varepsilon(U,V) \\
&= |a|^\alpha(1 - \sup_{U,V} \lambda_\varepsilon(U,V)) + \sup_{U,V} \lambda_\varepsilon(U,V) \qquad (21) \\
&\doteq \Delta_\varepsilon(a,\alpha)
\end{aligned}
$$

where the upper bound $\sup_{U,V} \Delta_\varepsilon^{U,V}(a,\alpha)$ has an interpolation form, which can be viewed as the convex combination between $|a|^\alpha$ and 1 with the coefficient $\sup_{U,V} \lambda_\varepsilon(U,V)$ determined by the probability measure set. More importantly, $\sup_{U,V} \Delta_\varepsilon^{U,V}(a,\alpha)$ is strictly less than 1, which is guaranteed by the finite set of $\{\lambda_\varepsilon(U,V)\}$ thanks to a finite set of interested probability measures. Finally, we have the variant of scale-sensitive property as follows, where the factor $\Delta_\varepsilon(a,\alpha)$ depends on $\alpha, a$ and the probability measure set.

$$
\mathcal{W}_{c,\varepsilon}(aU, aV) \leq \Delta_\varepsilon(a,\alpha)\mathcal{W}_{c,\varepsilon}(U,V). \qquad (22)
$$

## E  Proof of Theorem 1

### E.1  $\varepsilon \to 0$ and $c = -k_\alpha$.

We study the uniform convergence when $\varepsilon \to 0$. The proof is summarized from the optimal transport literature [25, 21] and we here provide the detailed proof for completeness. On the one hand, $\mathcal{W}_{c,\varepsilon} \geq \int(x-y)^\alpha d\Pi^*(x,y)dxdy \geq W_\alpha^\alpha$ as KL $\geq 0$. We want to provide the inequality on the other side. Denote $\Pi'$ as the minimizer in the Wasserstein distance $W_\alpha^\alpha$. For any $\delta > 0$, there always exists a joint distribution $\Pi^\delta$ such that

$$
| \int(x-y)^\alpha d\Pi'(x,y) - \int(x-y)^\alpha d\Pi^\delta(x,y)| \leq \delta \qquad (23)
$$

and KL$(\Pi^\delta|\mu \otimes \nu) < +\infty$, i.e., $\int(x-y)^\alpha d\Pi^\delta(x,y) - \int(x-y)^\alpha d\Pi'(x,y) \leq \delta$. One possible way to find $\Pi^\delta$ is provided in notes of Lecture 6 in Optimal Transport Course[4] and we invite interested readers for reference. It follows that

$$
W_\alpha^\alpha \leq \mathcal{W}_{c,\varepsilon} \leq \int(x-y)^\alpha d\Pi^\delta(x,y) + \varepsilon\text{KL}(\Pi^\delta|\mu \otimes \nu) \leq \int(x-y)^\alpha d\Pi'(x,y) + \delta + \varepsilon\text{KL}(\Pi^\delta|\mu \otimes \nu),
$$
$$(24)$$

where the RHS $\int(x-y)^\alpha d\Pi'(x,y) + \delta + \varepsilon\text{KL}(\Pi^\delta|\mu \otimes \nu) \to \int(x-y)^\alpha d\Pi'(x,y) + \delta = W_\alpha^\alpha + \delta$ as $\varepsilon \to 0$. As $\delta > 0$ is arbitrary, combing the two sides, it shows that $\mathcal{W}_{c,\epsilon} \to W_\alpha^\alpha$ as $\varepsilon \to 0$. Thus, Sinkhorn divergence maintains the properties of Wasserstein distance when $\varepsilon \to 0$.

When $\varepsilon = 0$, it has been shown that $W_\alpha$ can guarantee a $\gamma$-contraction property for distributional Bellman operator [5]. The crux of proof is that $W_\alpha$ is $\gamma$-scale sensitive:

$$
\begin{aligned}
W_\alpha(aU, aV) &= \left( \inf_{\Pi \in \Pi(aU,aV)} \int a^\alpha(x-y)^p d\Pi(x,y) \right)^{1/\alpha} \\
&\leq a \left( \inf_{\Pi \in \Pi(U,V)} \int (x-y)^p d\Pi(x,y) \right)^{1/\alpha} \qquad (25) \\
&= aW_\alpha(U,V),
\end{aligned}
$$

where the inequality comes from the change of optimal joint distribution. Therefore, $W_\alpha(aU, aV) \leq aW_\alpha(U,V)$ guarantees a $\gamma$-contraction property for the distributional Bellman operator. As such, for $W_\alpha^\alpha$, when $\varepsilon = 0$, it suggest that $\overline{\mathcal{W}}_{c,0} = W_\alpha^\alpha$ corresponds to a $\gamma^\alpha$-contraction for the distributional Bellman operator $\mathfrak{T}^\pi$.

---

[4] `https://lchizat.github.io/ot2021orsay.html`

## E.2 $\varepsilon \to \infty$ and $c = -k_\alpha$.

Our complete proof is inspired by [40, 25]. Recap the definition of squared MMD is

$$\mathbb{E}\left[k\left(\mathbf{X}, \mathbf{X}'\right)\right] + \mathbb{E}\left[k\left(\mathbf{Y}, \mathbf{Y}'\right)\right] - 2\mathbb{E}[k(\mathbf{X}, \mathbf{Y})]. \tag{26}$$

When the kernel function $k$ degenerates to an unrectified $k_\alpha(x, y) := -\|x - y\|^\alpha$ for $\alpha \in (0, 2)$, the squared MMD would degenerate to

$$2\mathbb{E}\|\mathbf{X} - \mathbf{Y}\|^\alpha - \mathbb{E}\|\mathbf{X} - \mathbf{X}'\|^\alpha - \mathbb{E}\|\mathbf{Y} - \mathbf{Y}'\|^\alpha. \tag{27}$$

where $\mathbf{X}, \mathbf{X}' \overset{\text{i.i.d.}}{\sim} \mu, \mathbf{Y}, \mathbf{Y}' \overset{\text{i.i.d.}}{\sim} \nu$ and $\mathbf{X}, \mathbf{X}', \mathbf{Y}, \mathbf{Y}'$ are mutually independent. On the other hand, by definition, we have the Sinkhorn loss as

$$\overline{\mathcal{W}}_{c,\infty}(\mu, \nu) = 2\mathcal{W}_{c,\infty}(\mu, \nu) - \mathcal{W}_{c,\infty}(\mu, \mu) - \mathcal{W}_{c,\infty}(\nu, \nu). \tag{28}$$

Denoting $\Pi_\varepsilon$ be the unique minimizer for $\overline{\mathcal{W}}_{c,\varepsilon}$, it holds that $\Pi_\varepsilon \to \mu \otimes \nu$ as $\varepsilon \to \infty$, which is the product of two marginal distributions. That being said, $\mathcal{W}_{c,\infty}(\mu, \nu) \to \int c(x, y)\mathrm{d}\mu(x)\mathrm{d}\nu(y) + 0 = \int c(x, y)\mathrm{d}\mu(x)\mathrm{d}\nu(y)$. *One important proof insight here is although $\varepsilon \to +\infty$, the KL term tends to zero, which is faster than $\varepsilon$. Therefore, the whole regularization term still tends to 0 as $\varepsilon \to +\infty$.* If $c = -k_\alpha = \|x-y\|^\alpha$, we eventually have $\mathcal{W}_{-k_\alpha,\infty}(\mu, \nu) \to \int \|x-y\|^\alpha \mathrm{d}\mu(x)\mathrm{d}\nu(y) = \mathbb{E}\|\mathbf{X}-\mathbf{Y}\|^\alpha$, where $\mu$ and $\nu$ can be inherently correlated, although the minimizer degenerates to the product of the two marginal distributions. Finally, we can have

$$\overline{\mathcal{W}}_{-k_\alpha,\infty} \to 2\mathbb{E}\|\mathbf{X} - \mathbf{Y}\|^\alpha - \mathbb{E}\|\mathbf{X} - \mathbf{X}'\|^\alpha - \mathbb{E}\|\mathbf{Y} - \mathbf{Y}'\|^\alpha, \tag{29}$$

which is exactly the form of squared MMD with the unrectified kernel $k_\alpha$. Now the key is to prove that $\Pi_\varepsilon \to \mu \otimes \nu$ as $\varepsilon \to \infty$. We give the detailed proof as follows.

Firstly, it is apparent that $\mathcal{W}_{c,\varepsilon}(\mu, \nu) \leq \int c(x, y)\mathrm{d}\mu(x)\mathrm{d}\nu(y)$ as $\mu \otimes \nu \in \Pi(\mu, \nu)$. Let $\{\varepsilon_k\}$ be a positive sequence that diverges to $\infty$, and $\Pi_k$ be the corresponding sequence of unique minimizers for $\mathcal{W}_{c,\varepsilon}$. According to the optimality condition, it must be the case that $\int c(x, y)\mathrm{d}\Pi_k + \varepsilon_k \mathrm{KL}(\Pi_k, \mu \otimes \nu) \leq \int c(x, y)\mathrm{d}\mu \otimes \nu + 0$ (when $\Pi(\mu, \nu) = \mu \otimes \nu$). Thus,

$$\mathrm{KL}\left(\Pi_k, \mu \otimes \nu\right) \leqslant \frac{1}{\varepsilon_k}\left(\int c \, \mathrm{d}\mu \otimes \nu - \int c \, \mathrm{d}\Pi_k\right) \to 0.$$

Besides, by the compactness of $\Pi(\mu, \nu)$, we can extract a converging subsequence $\Pi_{n_k} \to \Pi_\infty$. Since KL is weakly lower-semicontinuous, it holds that

$$\mathrm{KL}\left(\Pi_\infty, \mu \otimes \nu\right) \leqslant \lim_{k \to \infty} \inf \mathrm{KL}\left(\Pi_{n_k}, \mu \otimes \nu\right) = 0$$

Hence $\Pi_\infty = \mu \otimes \nu$. That being said that the optimal coupling is simply the product of the marginals, indicating that $\Pi_\varepsilon \to \mu \otimes \nu$ as $\varepsilon \to \infty$. As a special case, when $\alpha = 1$, $\overline{\mathcal{W}}_{-k_1,\infty}(u, v)$ is equivalent to the energy distance

$$d_E(\mathbf{X}, \mathbf{Y}) := 2\mathbb{E}\|\mathbf{X} - \mathbf{Y}\| - \mathbb{E}\|\mathbf{X} - \mathbf{X}'\| - \mathbb{E}\|\mathbf{Y} - \mathbf{Y}'\|. \tag{30}$$

In summary, if the cost function is the rectified kernel $k_\alpha$, it is the case that $\overline{\mathcal{W}}_{-k_\alpha,\varepsilon}$ converges to the squared MMD as $\varepsilon \to \infty$. According to [37], $\mathfrak{T}^\pi$ is $\gamma^{\alpha/2}$-contractive in the supremal form of MMD with the unrectified kernel $k_\alpha$. As $\overline{\mathcal{W}}_{c,\varepsilon}(\mu, \nu) \to \mathrm{MMD}_{k_\alpha}^2(\mu, \nu)$, which is a squared MMD instead of MMD, it implies that $\mathfrak{T}^\pi$ is $\gamma^\alpha$-contractive under the squared MMD / $\overline{\mathcal{W}}_{c,+\infty}$.

## E.3 $\varepsilon \in (0, +\infty)$ and $c = -\kappa_\alpha$

In the proof of Proposition 2, we have shown that the Sinkhorn loss $\mathcal{W}_{c,\varepsilon}$ satisfies the sum-invariant (**I**) and a new variant of scale-sensitive properties as follows:

$$\mathcal{W}_{c,\varepsilon}(A + U, A + V) \leq \mathcal{W}_{c,\varepsilon}(U, V)$$
$$\mathcal{W}_{c,\varepsilon}(aU, aV) \leq \Delta_\varepsilon(a, \alpha)\mathcal{W}_{c,\varepsilon}(U, V). \tag{31}$$

The Sinkhorn divergence $\overline{\mathcal{W}}_{c,\varepsilon}$ is defined by additionally subtracting two self-distance terms ($\mathcal{W}_{c,\varepsilon}(\mu, \mu)$ and $\mathcal{W}_{c,\varepsilon}(\nu, \nu)$) based on $\mathcal{W}_{c,\varepsilon}(\mu, \nu)$ in order to guarantee the non-negativity, triangularity and metric properties. These two self-distance terms do not change the (**I**) and (**S**)

properties when extending $\mathcal{W}_{c,\varepsilon}$ to $\overline{\mathcal{W}}_{c,\varepsilon}$, and some proof techniques can refer to Section 2 in [21]. The only difference is that the scaling factor will be $\overline{\Delta}_{\varepsilon}^{U,V}(a,\alpha)$, which is the counterpart of Eq. 20 satisfying

$$\overline{\mathcal{W}}_{c,\varepsilon}(aU, aV) \leq \overline{\Delta}_{\varepsilon}^{U,V}(a,\alpha)\overline{\mathcal{W}}_{c,\varepsilon}(U,V). \tag{32}$$

where $\overline{\Delta}_{\varepsilon}^{U,V}(a,\alpha) = |a|^{\alpha}(1 - \overline{\lambda}_{\varepsilon}(U,V)) + \overline{\lambda}_{\varepsilon}(U,V) \in (|a|^{\alpha}, 1)$ for $\varepsilon \in (0, +\infty)$ and $a < 1$ due to the fact that $\overline{\lambda}_{\varepsilon}(U,V) \in (0,1)$ for any non-trivial $\overline{\mathcal{W}}_{c,\varepsilon}(U,V)$. The new ratio $\overline{\lambda}_{\varepsilon}(U,V) = \frac{\varepsilon \text{KL}(\Pi^*|\mu\otimes\nu)}{\overline{\mathcal{W}}_{c,\varepsilon}} \in (0,1)$ for any considered $U, V$ with measures $\mu, \nu$ in the interested probability measure set. In particular, in the context of distributional RL, the set of interested probability measures would be the return distribution set of $\{Z(s,a)\}$ for $s \in \mathcal{S}$ and $a \in \mathcal{A}$ in a given finite MDP. We now want to find the universal upper bound $\overline{\Delta}_{\varepsilon}(a,\alpha)$, which is defined as

$$\overline{\Delta}_{\varepsilon}(a,\alpha) = |a|^{\alpha}(1 - \sup_{U,V}\overline{\lambda}_{\varepsilon}(U,V)) + \sup_{U,V}\overline{\lambda}_{\varepsilon}(U,V) \in (|a|^{\alpha}, 1). \tag{33}$$

Following the proof in Appendix D, the finite MDP guarantees a finite ratio set of $\{\overline{\lambda}_{\varepsilon}(U,V)\}$, and thus we can find a universal upper bound $\overline{\lambda}^*$ of the ratio set such that $\{\overline{\lambda}_{\varepsilon}(U,V)\} \leq \overline{\lambda}^* < 1$. This also implies that $\sup_{U,V}\overline{\lambda}_{\varepsilon}(U,V) \in (0,1)$ and thus the scaling factor $\overline{\Delta}_{\varepsilon}(a,\alpha) \in (|a|^{\alpha}, 1)$, which is strictly less than 1. Therefore, we have the **(I)** and **(S)** properties of $\overline{\mathcal{W}}_{c,\varepsilon}$:

$$\overline{\mathcal{W}}_{c,\varepsilon}(A + U, A + V) \leq \overline{\mathcal{W}}_{c,\varepsilon}(U,V)$$
$$\overline{\mathcal{W}}_{c,\varepsilon}(aU, aV) \leq \overline{\Delta}_{\varepsilon}(a,\alpha)\overline{\mathcal{W}}_{c,\varepsilon}(U,V). \tag{34}$$

Putting all together, we now derive the convergence of distributional Bellman operator $\mathfrak{T}^{\pi}$ under the supreme form of $\overline{\mathcal{W}}_{c,\varepsilon}$, i.e., $\overline{\mathcal{W}}_{c,\varepsilon}^{\infty}$:

$$
\begin{aligned}
\overline{\mathcal{W}}_{c,\varepsilon}^{\infty}(\mathfrak{T}^{\pi}Z_1, \mathfrak{T}^{\pi}Z_2) &= \sup_{s,a}\overline{\mathcal{W}}_{c,\varepsilon}(\mathfrak{T}^{\pi}Z_1(s,a), \mathfrak{T}^{\pi}Z_2(s,a)) \\
&= \sup_{s,a}\overline{\mathcal{W}}_{c,\varepsilon}(R(s,a) + \gamma Z_1(s',a'), R(s,a) + \gamma Z_2(s',a')) \\
&\overset{(a)}{\leq} \overline{\mathcal{W}}_{c,\varepsilon}(\gamma Z_1(s',a'), \gamma Z_2(s',a')) \\
&\overset{(b)}{\leq} \overline{\Delta}_{\varepsilon}^{Z_1(s',a'),Z_2(s',a')}(\gamma,\alpha)\overline{\mathcal{W}}_{c,\varepsilon}(Z_1(s',a'), Z_2(s',a')) \\
&\leq \sup_{s',a'}\overline{\Delta}_{\varepsilon}^{Z_1(s',a'),Z_2(s',a')}(\gamma,\alpha)\sup_{s',a'}\overline{\mathcal{W}}_{c,\varepsilon}(Z_1(s',a'), Z_2(s',a')) \\
&= \overline{\Delta}_{\varepsilon}(\gamma,\alpha)\overline{\mathcal{W}}_{c,\varepsilon}^{\infty}(Z_1, Z_2)
\end{aligned}
\tag{35}
$$

where the inequality (a) is based on the sum invariant property **(I)** of Sinkhorn divergence. (b) is based on the new variant of scale-sensitive property **(S)** of Sinkhorn divergence and the leverage of $c = -k_{\alpha}$. Notably, $\overline{\Delta}_{\varepsilon}(\gamma,\alpha) \in (|\gamma|^{\alpha}, 1)$ is an MDP-dependent constant (determined by the return distribution set), which is also determined by $\gamma, \varepsilon$ and $\alpha$. As such, we conclude that distributional Bellman operator is *at least* $\overline{\Delta}_{\varepsilon}(\gamma,\alpha)$-contractive, where the contraction factor $\overline{\Delta}_{\varepsilon}(\gamma,\alpha)$ is strictly less than 1 in a given finite MDP. Based on the existing Banach fixed point theorem, we have a unique optimal return distribution by applying the distributional Bellman operator $\mathfrak{T}^{\pi}$ in the distributional dynamic programming when convergence.

# F  Proof of Corollary 1

*Proof.* The contraction conclusion that extends to the multi-dimensional return distributions is straightforward. As the definition of Sinkhorn divergence inherently allows the multi-dimensional measures, the sum-invariant and the variant of scale-sensitive properties hold naturally. Specifically, after recapping to proof of these properties, we only need to change $c(x,y) = (x-y)^{\alpha}$ to $c(\mathbf{x},\mathbf{y}) = \|\mathbf{x} - \mathbf{y}\|^{\alpha}$ and re-define two $d$-dimensional random vector $\mathbf{U}$ and $\mathbf{V}$ with the $d$-dimensional probability measure $\mu$ and $\nu$. Therefore, the **(I)** and **(S)** properties in the multi-dimensional reward settings are:

$$\overline{\mathcal{W}}_{c,\varepsilon}(\mathbf{A} + \mathbf{U}, \mathbf{A} + \mathbf{V}) \leq \overline{\mathcal{W}}_{c,\varepsilon}(\mathbf{U},\mathbf{V})$$
$$\overline{\mathcal{W}}_{c,\varepsilon}(a\mathbf{U}, a\mathbf{V}) \leq \overline{\Delta}_{\varepsilon}(a,\alpha)\overline{\mathcal{W}}_{c,\varepsilon}(\mathbf{U},\mathbf{V}), \tag{36}$$

where $\mathbf{A}$ is a $d$-dimensional random vector independent of $\mathbf{U}$ and $\mathbf{V}$.

By leveraging these two properties, we now derive the convergence of distributional Bellman operator $\mathfrak{T}_d^\pi$ under $\overline{\mathcal{W}}_{c,\varepsilon}^\infty$ in the joint return distribution setting. Given two $d$-dimensional return distributions $\mathbf{Z}_1$ and $\mathbf{Z}_2$, we have

$$
\begin{aligned}
\overline{\mathcal{W}}_{c,\varepsilon}^\infty(\mathfrak{T}_d^\pi\mathbf{Z}_1, \mathfrak{T}_d^\pi\mathbf{Z}_2) &= \sup_{s,a} \overline{\mathcal{W}}_{c,\varepsilon}(\mathfrak{T}_d^\pi\mathbf{Z}_1(s,a), \mathfrak{T}_d^\pi\mathbf{Z}_2(s,a)) \\
&= \sup_{s,a} \overline{\mathcal{W}}_{c,\varepsilon}(\mathbf{R}(s,a) + \gamma\mathbf{Z}_1(s',a'), \mathbf{R}(s,a) + \gamma\mathbf{Z}_2(s',a')) \\
&\stackrel{(a)}{\leq} \overline{\mathcal{W}}_{c,\varepsilon}(\gamma\mathbf{Z}_1(s',a'), \gamma\mathbf{Z}_2(s',a')) \\
&\stackrel{(b)}{\leq} \overline{\Delta}_\varepsilon^{\mathbf{Z}_1(s',a'),\mathbf{Z}_2(s',a')}(\gamma,\alpha)\overline{\mathcal{W}}_{c,\varepsilon}(\mathbf{Z}_1(s',a'), \mathbf{Z}_2(s',a')) \\
&\leq \sup_{s',a'} \overline{\Delta}_\varepsilon^{\mathbf{Z}_1(s',a'),\mathbf{Z}_2(s',a')}(\gamma,\alpha) \sup_{s',a'} \overline{\mathcal{W}}_{c,\varepsilon}(\mathbf{Z}_1(s',a'), \mathbf{Z}_2(s',a')) \\
&= \overline{\Delta}_\varepsilon(\gamma,\alpha)\overline{\mathcal{W}}_{c,\varepsilon}^\infty(\mathbf{Z}_1, \mathbf{Z}_2)
\end{aligned}
\tag{37}
$$

where the inequality (a) is based on the sum invariant property **(I)** of Sinkhorn divergence that cancels the additive $d$-dimensional random vector $\mathbf{R}(s,a)$. (b) is based on the new variant of scale-sensitive property **(S)** of Sinkhorn divergence and the leverage of $c = -k_\alpha$, where the contraction factor $\overline{\Delta}_\varepsilon(\gamma,\alpha)$ will depend on the set of $d$-dimensional probability measures/distributions. Notably, the analysis of $\overline{\Delta}_\varepsilon(\gamma,\alpha)$ in the one-dimensional return setting established in Appendix D and Appendix E is also applicable in the multi-dimensional setting. $\qquad\square$

# G   Algorithm: Sinkhorn Iterations and Sinkhorn Distributional RL

---

**Algorithm 2** Sinkhorn Iterations to Approximate $\overline{\mathcal{W}}_{c,\varepsilon}\left(\{Z_i\}_{i=1}^N, \{\mathfrak{T}Z_j\}_{j=1}^N\right)$

---

**Input**: Two samples sequences $\{Z_i\}_{i=1}^N, \{\mathfrak{T}Z_j\}_{j=1}^N$, number of iterations $L$ and hyperparameter $\varepsilon$.

1: $\hat{c}_{i,j} = c(Z_i, \mathfrak{T}Z_j)$ for $\forall i = 1, ..., N, j = 1, ..., N$
2: $\mathcal{K}_{i,j} = \exp(-\hat{c}_{i,j}/\varepsilon)$
3: $b_0 \leftarrow \mathbf{1}_N$
4: **for** $l = 1, 2, ..., L$ **do**
5: $\quad a_l \leftarrow \frac{\mathbf{1}_N}{\mathcal{K}b_{l-1}}, b_l \leftarrow \frac{\mathbf{1}_N}{\mathcal{K}a_l}$
6: **end for**
7: $\widehat{\overline{\mathcal{W}}}_{c,\varepsilon}\left(\{Z_i\}_{i=1}^N, \{\mathfrak{T}Z_j\}_{j=1}^N\right) = \langle(K \odot \hat{c})b, a\rangle$

**Return**: $\widehat{\overline{\mathcal{W}}}_{c,\varepsilon}\left(\{Z_i\}_{i=1}^N, \{\mathfrak{T}Z_j\}_{j=1}^N\right)$

---

Given two sample sequences $\{Z_i\}_{i=1}^N, \{\mathfrak{T}Z_j\}_{j=1}^N$ in the distributional RL algorithm, the optimal transport distance is equivalent to the form:

$$
\min_{P\in\mathbb{R}_+^{N\times N}} \left\{\langle P, \hat{c}\rangle; P\mathbf{1}_N = \mathbf{1}_N, P^\top\mathbf{1}_N = \mathbf{1}_N\right\},
\tag{38}
$$

where the empirical cost function is $\hat{c}_{i,j} = c(Z_i, \mathfrak{T}Z_j)$. By adding entropic regularization on optimal transport distance, Sinkhorn divergence can be viewed to restrict the search space of $P$ in the following scaling form:

$$
P_{i,j} = a_i\mathcal{K}_{i,j}b_j,
\tag{39}
$$

where $\mathcal{K}_{i,j} = e^{-\hat{c}_{i,j}/\varepsilon}$ is the Gibbs kernel defined in Eq. 2. This allows us to leverage iterations regarding the vectors $a$ and $b$. More specifically, we initialize $b_0 = \mathbf{1}_N$, and then the Sinkhorn iterations are expressed as

$$
a_{l+1} \leftarrow \frac{\mathbf{1}_N}{\mathcal{K}b_l} \quad \text{and} \quad b_{l+1} \leftarrow \frac{\mathbf{1}_N}{\mathcal{K}^\top a_{l+1}},
\tag{40}
$$

where $\dot{\div}$ indicates an entry-wise division. Combining Sinkhorn Iteration in Algorithm 2 and the generic update of Sinkhorn Distributional RL in Algorithm 1, we provide a full version of Sinkhorn Distributional RL algorithm in Algorithm 3.

---

**Algorithm 3** Sinkhorn Distributional RL

---

**Require:** Number of generated samples $N$, the kernel $k$ (e.g., unrectified kernel), discount factor $\gamma \in [0, 1]$, learning rate $\alpha$, replay buffer $M$, main network $Z_\theta$, target network $Z_{\theta^*}$, number of iterations $L$, hyperparameter $\varepsilon$, and a behavior policy $\pi$ based on $Z_\theta$ following an $\epsilon$-greedy rule
1: Initialize $\theta$ and $\theta^* \leftarrow \theta$
2: **for** $t = 1, 2, \ldots$ **do**
3:     Take action $a_t \sim \pi(\cdot|s_t; \theta)$, receive reward $r_t \sim R(\cdot|s_t, a_t)$, and observe $s_{t+1} \sim P(\cdot|s_t, a_t)$
4:     Store $(s_t, a_t, r_t, s_{t+1})$ to the replay buffer $M$
5:     Randomly draw a batch of transition samples $(s, a, r, s')$ from the replay buffer $M$
6:     Compute a greedy action: $a^* = \arg\max_{a' \in A} \frac{1}{N} \sum_{i=1}^{N} Z_{\theta^*}(s', a')_i$
7:     Compute the target Bellman return distribution: $\mathfrak{T}Z_i \leftarrow r + \gamma Z_{\theta^*}(s', a^*)_i, \forall 1 \leq i \leq N$
8:     Evaluate Sinkhorn divergence via Sinkhorn Iterations in Algorithm 2:

$$\overline{\mathcal{W}}_{c,\varepsilon}\left(\{Z_\theta(s,a)_i\}_{i=1}^{N}, \{\mathfrak{T}Z_j\}_{j=1}^{N}\right)$$

9:     Update the main network $Z_\theta$: $\theta \leftarrow \theta - \alpha \nabla_\theta \overline{\mathcal{W}}_{c,\varepsilon}\left(\{Z_\theta(s,a)_i\}_{i=1}^{N}, \{\mathfrak{T}Z_j\}_{j=1}^{N}\right)$
10:     Periodically update the target network $\theta^* \leftarrow \theta$
11: **end for**

---

# H  Summary Table for Human Normalized Scores (HNS)

|  | **Mean** | **IQM (5%)** | **Median** | **>DQN** |
|---|---|---|---|---|
| DQN | 452.6 % | 181.2 % | 32.8 % | 0 |
| C51 | 640.2 % | 368.5 % | 68.5 % | 35 |
| QR-DQN-1 | 780.9 % | 401.9 % | 85.8 % | 38 |
| MMD-DQN | 781.7 % | 428.4 % | **96.6 %** | 37 |
| SinkhornDRL | **1306.1 %** | **477.0 %** | 91.1 % | **41** |

Table 2: Evaluation of best Human Normalized Scores (HNS) across 55 Atari games. Results are averaged over 3 seeds. Our proposed SinkhornDRL achieves the best performance in terms of Mean and IQM(5%) HNS as well as the "> DQN" metric, and is on par with MMD-DQN in terms of Median of HNS.

# I  Learning Curves on 55 Atari Games

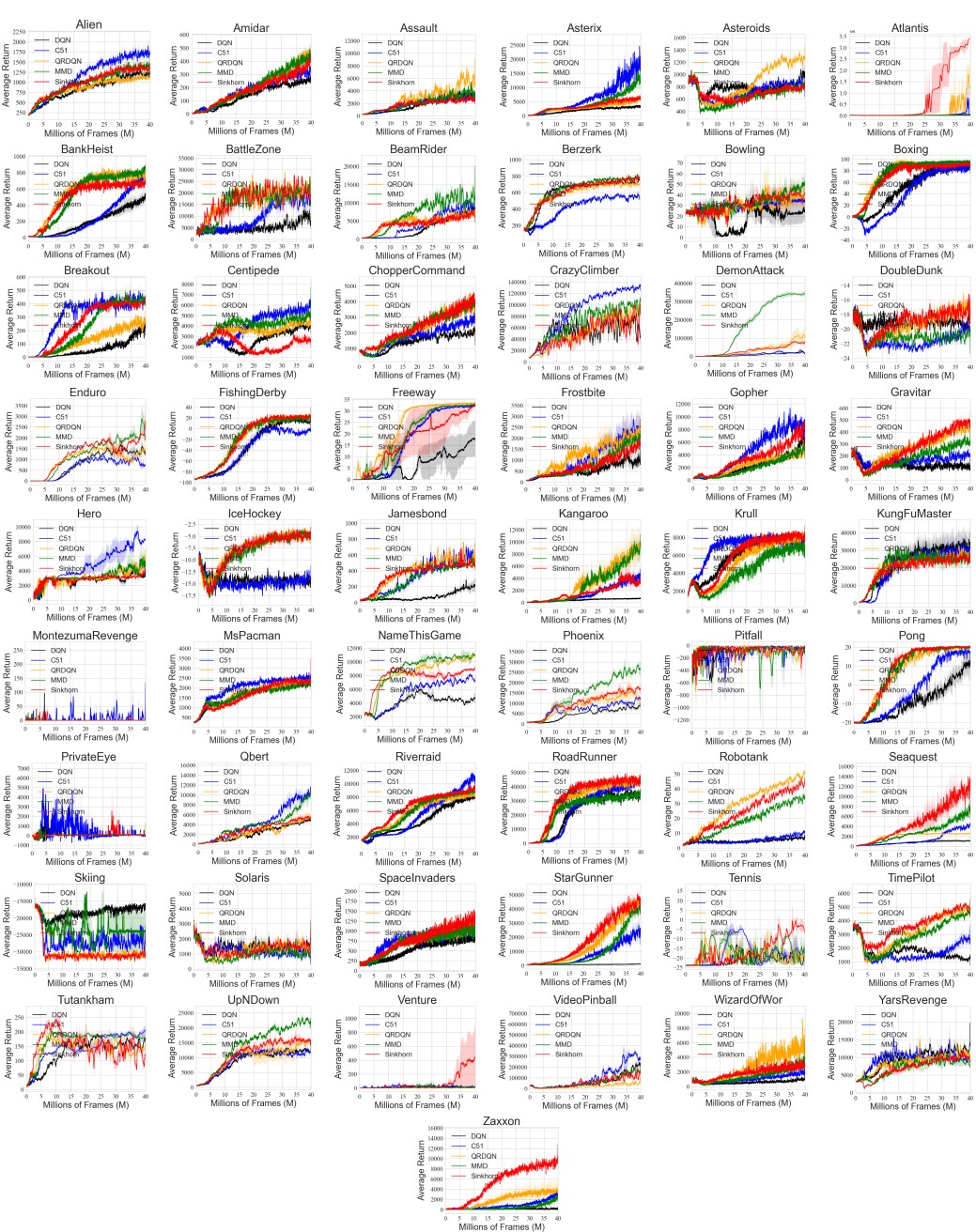

Figure 6: Learning curves of SinkhornDRL on 55 Atari games after training 40M frames averaged over 3 seeds.

## J  Raw Score Table Across 55 Atari Games

| GAMES | RANDOM | HUMAN | DQN | C51 | QR-DQN | MMD-DQN | SinkhornDRL |
|---|---|---|---|---|---|---|---|
| Alien | 211.9 | 7,127.7 | 1030 | 1510 | 1030 | 1480 | **1560** |
| Amidar | 2.34 | 1,719.5 | 341 | 424 | **677** | 510 | 588 |
| Assault | 283.5 | 742.0 | 3232 | 3647 | **12943** | 3295 | 2960 |
| Asterix | 268.5 | 8,503.3 | 3000 | **34900** | 11500 | 14900 | 6500 |
| Asteroids | 1008.6 | 47,388.7 | 1180 | 780 | **1650** | 1080 | 1370 |
| Atlantis | 22188 | 29,028.1 | 15500 | 84900 | 3316700 | 93600 | **3447100** |
| BankHeist | 14 | 753.1 | 570 | 960 | **980** | 880 | 700 |
| BattleZone | 3000 | 37,187.5 | 15000 | 19000 | 26000 | **35000** | 32000 |
| BeamRider | 414.3 | 16,926.5 | 8200 | 7476 | 7642 | **25602** | 6022 |
| Berzerk | 165.6 | 2,630.4 | **970** | 650 | 640 | 860 | 910 |
| Bowling | 23.48 | 160.7 | 54 | 43 | **60** | **60** | **60** |
| Boxing | -0.69 | 12.1 | 94 | 90 | **100** | **100** | **100** |
| Breakout | 1.5 | 30.5 | 343 | **452** | 414 | 432 | 418 |
| Centipede | 2064.77 | 12,017.0 | 7551 | 4133 | 5388 | **9342** | 4070 |
| ChopperCommand | 794 | 7,387.8 | 1500 | **3600** | 3500 | **3600** | 3400 |
| CrazyClimber | 8043 | 35,829.4 | 94300 | **153100** | 139500 | 98500 | 137400 |
| DemonAttack | 162.25 | 1,971.0 | 31420 | 50240 | 240660 | **407030** | 105185 |
| DoubleDunk | -18.14 | -16.4 | -16 | -20 | -18 | -22 | **-12** |
| Enduro | 0.01 | 860.5 | 1387 | 1086 | 1972 | 1953 | **4608** |
| FishingDerby | -93.06 | -38.7 | 23 | -1 | 31 | 31 | **61** |
| Freeway | 0.01 | 29.6 | 31 | 32 | **34** | 33 | **34** |
| Frostbite | 73.2 | 4,334.7 | 3330 | **3690** | 3470 | 3250 | 2640 |
| Gopher | 364 | 2,412.5 | 11400 | **14780** | 5440 | 3740 | 6620 |
| Gravitar | 226.5 | 3,351.4 | 350 | 350 | **750** | 350 | 500 |
| Hero | 551 | 30,826.4 | 3440 | 8535 | **10155** | 7195 | 6540 |
| IceHockey | -10.3 | 0.9 | -13 | -10 | -4 | -3 | **-2** |
| Jamesbond | 27 | 302.8 | 350 | 600 | **650** | 450 | 500 |
| Kangaroo | 54 | 3,035.0 | 1300 | 6500 | 14600 | **14800** | 3600 |
| Krull | 1,566.59 | 2,665.5 | 8892 | 9336 | **10053** | 7762 | 9630 |
| KungFuMaster | 451 | 22,736.3 | **46500** | 38000 | 27900 | 26900 | 43600 |
| MontezumaRevenge | 0.0 | 4,753.3 | 1 | **400** | 1 | 1 | 1 |
| MsPacman | 242.6 | 6,951.6 | 3230 | 2440 | 1860 | 3130 | **5120** |
| NameThisGame | 2404.9 | 8,049.0 | 6160 | 5750 | **13580** | 9350 | 11250 |
| Phoenix | 757.2 | 7,242.6 | 9430 | 18780 | 9390 | **25690** | 23300 |
| Pitfall | -265 | 6,463.7 | 1 | 1 | 1 | 1 | 1 |
| Pong | -20.34 | 14.6 | **21** | 20 | 20 | **21** | **21** |
| PrivateEye | 34.49 | 69,571.3 | 100 | 100 | 100 | 100 | 100 |
| Qbert | 188.75 | 13,455.0 | 7425 | **16375** | 7800 | 16225 | 7750 |
| RiverRaid | 1575.4 | 17,118.0 | 8470 | **13310** | 8710 | 9190 | 9530 |
| RoadRunner | 7 | 7,845.0 | 45500 | **60900** | 52500 | 45600 | 59500 |
| Robotank | 2.24 | 11.9 | 8 | 11 | **58** | 39 | 54 |
| Seaquest | 88.2 | 42,054.7 | 1740 | 5940 | 2640 | 7370 | **8350** |
| Skiing | -16267.9 | -4,336.9 | -13681 | -20495 | -29970 | **-8986** | -23455 |
| Solaris | 2346.6 | 12,326.7 | 1640 | 660 | 2200 | 3380 | **7720** |
| SpaceInvaders | 136.15 | 1,668.7 | 940 | **2480** | 1170 | 770 | 1200 |
| StarGunner | 631 | 10,250.0 | 1200 | 17200 | 52900 | 52500 | **57500** |
| Tennis | -23.92 | -8.3 | -23 | -1 | -7 | -8 | **5** |
| TimePilot | 3682 | 5,229.2 | 800 | 4100 | 4400 | **8000** | 4500 |
| Tutankham | 15.56 | 167.6 | 201 | 213 | **220** | 141 | 137 |
| UpNDown | 604.7 | 11,693.2 | 14560 | 18440 | 13710 | **27370** | 18910 |
| Venture | 0.0 | 1,187.5 | 1 | 1 | 1 | 1 | **700** |
| VideoPinball | 15720.98 | 17,667.9 | 155165 | **576843** | 189460 | 69175 | 347700 |
| WizardOfWor | 534 | 4,756.5 | 1400 | 2400 | **14300** | 11500 | 4300 |
| YarsRevenge | 3271.42 | 54,576.9 | **28048** | 7882 | 17729 | 7520 | 9120 |
| Zaxxon | 8 | 9,173.3 | 1 | 3900 | 9100 | 4300 | **19500** |
| **Number of Best** | | | 4 | 12 | 15 | 13 | **17** |
| **Number of Second Best** | | | 6 | 7 | 10 | 8 | **16** |

Table 3: Best score of all algorithms over 3 seeds across 55 Atari games after training 40M Frames. **Bold** denotes the best performance, while the underline represents the second best performance. The number of games with the best and second best performance substantiate the superiority of our SinkhornDRL across all considered baseline algorithms.

## K Features of Atari Games

| GAMES | Action Space | Dynamics |
|---|---|---|
| Alien | 18 | Complex |
| Amidar | 6 | Simple |
| Assault | 7 | Complex |
| Asterix | 18 | Complex |
| Asteroids | 4 | Simple |
| Atlantis | 4 | Simple |
| BankHeist | 18 | Simple |
| BattleZone | 18 | Simple |
| BeamRider | 18 | Complex |
| Berzerk | 18 | Complex |
| Bowling | Continuous | Simple |
| Boxing | 6 | Simple |
| Breakout | 4 | Simple |
| Centipede | 18 | Complex |
| ChopperCommand | Continuous | Complex |
| CrazyClimber | 18 | Complex |
| DemonAttack | 18 | Complex |
| DoubleDunk | 18 | Simple |
| Enduro | 9 | Simple |
| FishingDerby | 18 | Simple |
| Freeway | 3 | Simple |
| Frostbite | 18 | Complex |
| Gopher | 18 | Simple |
| Gravitar | Continuous | Complex |
| Hero | 18 | Simple |
| IceHockey | Continuous | Simple |
| Jamesbond | 18 | Complex |
| Kangaroo | 18 | Complex |
| Krull | 18 | Complex |
| KungFuMaster | 18 | Complex |
| MontezumaRevenge | 18 | Complex |
| MsPacman | 9 | Simple |
| NameThisGame | 18 | Complex |
| Phoenix | 18 | Complex |
| Pitfall | 18 | Complex |
| Pong | 3 | Simple |
| PrivateEye | 18 | Complex |
| Qbert | 6 | Complex |
| Riverraid | 18 | Complex |
| RoadRunner | 18 | Simple |
| Robotank | 9 | Simple |
| Seaquest | 18 | Complex |
| Skiing | 9 | Simple |
| Solaris | 18 | Complex |
| SpaceInvaders | 6 | Simple |
| StarGunner | 18 | Complex |
| Tennis | 18 | Simple |
| TimePilot | 18 | Complex |
| Tutankham | 18 | Complex |
| UpNDown | 18 | Complex |
| Venture | 18 | Complex |
| VideoPinball | 6 | Simple |
| WizardOfWor | 12 | Complex |
| YarsRevenge | 18 | Complex |
| Zaxxon | 18 | Complex |

Table 4: Number of Action space and difficulty of environmental dynamics of 55 Atari games.

# L  Sensitivity Analysis and Computational Cost

## L.1  More results in Sensitivity Analysis

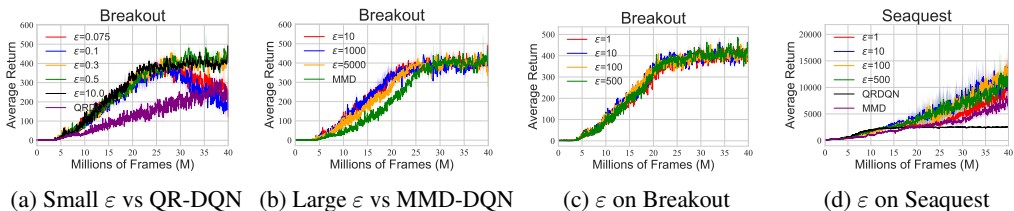

(a) Small $\varepsilon$ vs QR-DQN    (b) Large $\varepsilon$ vs MMD-DQN    (c) $\varepsilon$ on Breakout    (d) $\varepsilon$ on Seaquest

Figure 7: (a) Sensitivity analysis w.r.t. a small level of $\varepsilon$ SinkhornDRL to compare with QR-DQN that approximates Wasserstein distance on Breakout. (b) Sensitivity analysis w.r.t. a large level of $\varepsilon$ SinkhornDRL algorithm to compare with MMD-DQN on Breakout. All learning curves are reported over 2 seeds. (c) and (d) are results for a general $\varepsilon$ on Breakout and Seaquest, respectively.

**Decreasing $\varepsilon$.**  We argue that the limit behavior connection as stated in Theorem 1 may not be able to be verified rigorously via numeral experiments due to the numerical instability of Sinkhorn Iteration in Algorithm 2. From Figure 7 (a), we can observe that if we gradually decline $\varepsilon$ to 0, SinkhornDRL's performance tends to degrade and approach QR-DQN. Note that an overly small $\varepsilon$ will lead to a trivial almost 0 $\mathcal{K}_{i,j}$ in Sinkhorn iteration in Algorithm 2, and will cause $\frac{1}{0}$ numerical instability issue for $a_l$ and $b_l$ in Line 5 of Algorithm 2. In addition, we also conducted experiments on Seaquest, a similar result is also observed in Figure 7 (d). As shown in Figure 7 (d), the performance of SinkhornDRL is robust when $\varepsilon = 10, 100, 500$, but a small $\epsilon = 1$ tends to worsen the performance.

**Increasing $\varepsilon$.**  Moreover, for breakout, if we increase $\varepsilon$, the performance of SinkhornDRL tends to degrade and be close to MMD-DQN as suggested in Figure 7 (b). It is also noted that an overly large $\varepsilon$ will let the $\mathcal{K}_{i,j}$ explode to $\infty$. This also leads to the numerical instability issue in Sinkhorn iteration in Algorithm 2.

**Samples $N$.**  We find that SinkhornDRL requires a proper number of samples $N$ to perform favorably, and the sensitivity w.r.t $N$ depends on the environment. As suggested in Figure 8 (a), a smaller $N$, e.g., $N = 2$ on breakout has already achieved favorable performance and even accelerates the convergence in the early phase, while $N = 2$ on Seaquest will lead to the divergence issue. Meanwhile, an overly large $N$ worsens the performance across two games. We conjecture that using larger network networks to generate more samples may suffer from the overfitting issue, yielding the training instability [7]. In practice, we choose a proper number of samples, i.e., $N = 200$ across all games.

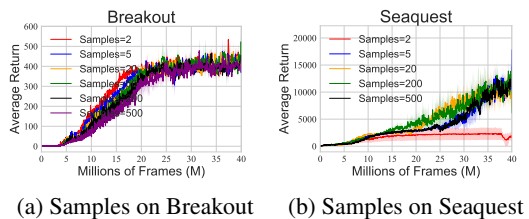

(a) Samples on Breakout    (b) Samples on Seaquest

Figure 8: Sensitivity analysis of Sinkhorn in terms of the number of samples $N$ on Breakout (a) and Seaquest (b).

**More Games on StarGunner and Zaxxon.**  Beyond Breakout and Seaquest, we also provide sensitivity analysis on StarGunner and Zaxxon games in Figure 9. It suggests overly small samples, e.g., 1 and overall large samples tend to degrade the performance, especially on Zaxxon. Although the two games are robust to $\varepsilon$, and we find a small or large $\varepsilon$ hurts the performance in Seaquest. Thus, considering all games, we set samples 200, and $\varepsilon = 10.0$ in a moderate range across all games, although a more careful tuning in each game will improve the performance further.

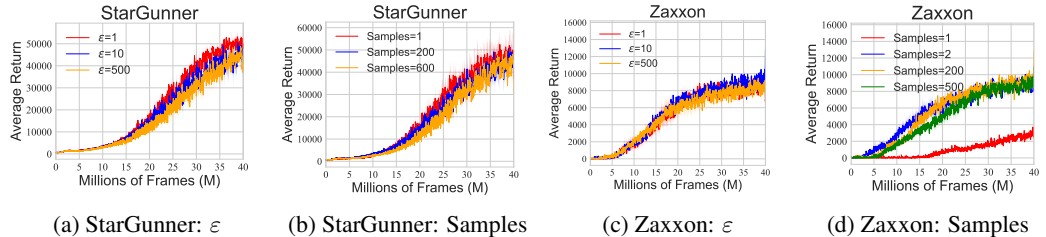

| (a) StarGunner: $\varepsilon$ | (b) StarGunner: Samples | (c) Zaxxon: $\varepsilon$ | (d) Zaxxon: Samples |

Figure 9: Sensitivity analysis of SinkhornDRL on StarGunner and Zaxxon in terms of $\varepsilon$, and number of samples. Learning curves are reported over 3 seeds.

## L.2 Comparison with the Computational Cost

We evaluate the computational time every 10,000 iterations across the whole training process of all considered distributional RL algorithms and make a comparison in Figure 10. It suggests that SinkhornDRL indeed increases around 50% computation cost compared with QR-DQN and C51, but only slightly increases the cost in contrast to MMD-DQN on both Breakout and Qbert games. We argue that this additional computational burden can be tolerant given the significant outperformance of SinkhornDRL in a large number of environments.

In addition, we also find that the number of Sinkhorn iterations $L$ is negligible to the computation cost, while an overly large sample $N$, e.g., 500, will lead to a large computational burden as illustrated in Figure 11. This can be intuitively explained as the computation complexity of the cost function $c_{i,j}$ is $\mathcal{O}(N^2)$ in SinkhornDRL, which is particularly heavy in the computation if $N$ is large enough.

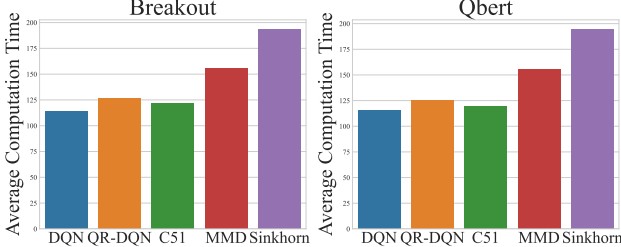

Figure 10: Average computational cost per 10,000 iterations of all considered distributional RL algorithm, where we select $\varepsilon = 10$, $L = 10$ and the number of samples $N = 200$ in SinkhornDRL algorithm.

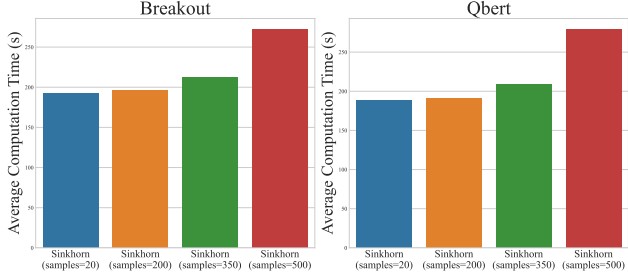

Figure 11: Average computational cost per 10,000 iterations of SinkhornDRL algorithm over different samples.

# M    Experimental Setting in Multi-dimensional Return Distributions

**Reward Structure and Decomposition.**    In practice, the reward function can be multi-dimensional [50, 30, 32, 15, 57, 31], where distributional RL is aimed at modeling multivariate return distribution with multiple reward sources. We follow the multi-dimensional return distribution setting in [57], which construct six Atari games with multiple sources of rewards by decomposing the scalar-valued primitive rewards into multi-dimension. For completeness, we introduce the respective reward structure and the decomposing method of the six considered Atari games, including AirRaid, Asteroids, Gopher, MsPacman, UpNDown, and Pong. The reward is decomposed while keeping the total reward unchanged.

- **AirRaid.** For primitive rewards, the agent kills different kinds of monsters and then receive discrete values of the rewards. The scalar-based primitive rewards are decomposed into four dimensions. The agent will get multi-dimensional rewards [100, 0, 0, 0], [0, 75, 0, 0], [0, 0, 50, 0],[0, 0, 0, 25], [0, 0, 0, 0] respectively for the primitive reward 100, 75, 50, 25 and 0.

- **Asteroids.** For primitive rewards, the agent kills different kinds of monsters and then receive values of the rewards. We denote the primitive reward as $r$, and decompose it into the three-dimensional reward as $[r_1, r_2, r_3]$. If $(r - 20) \bmod 50 = 0$, we let $r_1 = 20$, otherwise $r_1 = 0$. If $(r - r_1 - 50) \bmod 100 = 0$, we let $r_2 = 50$, otherwise $r_2 = 0$. We let $r_3 = r - r_1 - r_2$.

- **Gopher.** For primitive rewards, the agent gets $+80$ reward for killing a monster and $+20$ reward after removing holes on the ground. We denote the primitive reward as $r$, and decompose it into the two-dimensions as $[r_1, r_2, ]$. If $(r - 20) \bmod 100 = 0$, we let $r_1 = 20$, otherwise $r_1 = 0$. We let $r_2 = r - r_1$.

- **MsPacman.** For primitive rewards, the agent gets $\{+200, +400, +800, +1,600\}$ rewards after killing different monsters and $+10$ rewards after eating beans. In the reward decomposition, we decompose primitive reward denoted as $r$ into four dimensions $[r_1, r_2, r_3, r_4]$. If $(r - 10) \bmod 50 = 0$, we let $r_1 = 10$, otherwise $r_1 = 0$. If $(r - r_1 - 50) \bmod 100 = 0$, we let $r_2 = 50$, otherwise $r_2 = 0$. If $(r - r_1 - r_2 - 100) \bmod 200 = 0$, we let $r_3 = 100$, otherwise $r_3 = 0$. We let $r_4 = r - r_1 - r_2 - r_3$.

- **Pong.** For primitive rewards, the agent gets $+1$ if it wins a round, and $-1$ for losing the round. We decompose the reward into two-dimension: the agent will get $[-1, 0]$ for a $-1$ reward, $[0, 1]$ for a $+1$ reward; otherwise, $[0, 0]$.

- **UpNDown.** For primitive rewards, the agent gets $+400$ reward for killing an energy car, $+100$ for reaching a flag, and $+10$ reward for being alive. We denote the primitive reward as $r$, and decompose it into the three-dimensional reward as $[r_1, r_2, r_3]$. If $(r - 10) \bmod 100 = 0$, we let $r_1 = 10$, otherwise $r_1 = 0$. If $(r - r_1 - 100) \bmod 200 = 0$, we let $r_2 = 100$, otherwise $r_2 = 0$. We let $r_3 = r - r_1 - r_2$.

**Detailed Experimental Setup.** Our implementation extends our code in one-dimensional return setting to multi-dimensional return scenario and adopts the key aspects in [57]. For instance, similar to [57], we leverage a clipping reward normalizer to clip the multi-dimensional rewards into $[-1, 1]$ after applying the reward decomposition procedure mentioned above to the primitive rewards. We keep the same model architecture except only modifying the output of the last layer from $(B, |\mathcal{A}|, N)$ to $(B, |\mathcal{A}|, D, N)$, where $B$ is the batch size within each batch training, and $D$ is the dimension of the decomposed mutivariate reward function in each game.

**Baseline Algorithms.** Quantile regression can be used to approximate 1-Wasserstein distance in one-dimensional setting [14] as the one-dimensional Wassertein distance has a closed-form expression via the quantile function. However, it remains elusive how to use quantile regression to approximate multi-dimensional Wasserstein distance. This is to say, it is still unclear how to extend the quantile regression distributional RL (QR-DQN) into multi-dimensional return distribution setting, resulting in no proper baseline in our experiment. Despite that, we directly compare SinkrhornDRL with MMD-DQN [37] as MMD is applicable and computationally tractable in the multi-dimensional setting. Notably, we did not introduce other baselines, such as Hybrid Reward Architecture (HRA) [50], or MD3QN [57]. This is because 1) [57] shows that their proposed MD3QN and HRA do not outperform MMD-DQN in most of the six Atari games. By contrast, as suggested in Figure 4, our SinkhornDRL has already surpassed MMD-DQN across almost all the considered games, and thus excels over

MD3QN and HRA, correspondingly. 2) The primary focus of our study is the comprehensive advantages of SinkhornDRL over other distributional RL classes, especially in the more common setting within one-dimensional return distributions. The extension capability of SinkhornDRL into the multi-dimensional reward setting is one of its merits, which is not the primary focus of our study.

