# OpenReview forum: "Distributional Reinforcement Learning with Regularized Wasserstein Loss"
_NeurIPS.cc/2024/Conference — NeurIPS 2024 poster_

### Official Review · Reviewer_uu9w · 2024-07-08

**Soundness:** 3
**Presentation:** 3
**Contribution:** 2
**Rating:** 5
**Confidence:** 3

**Summary:**

This paper proposes a new RL algorithm that leverages Sinkhorn divergence, which they claimed as a regularized Wasserstein loss. Theoretically, they showed the contraction properties that align with the interpolation nature of Sinkhorn divergence between Wasserstein distance and MMD. Empirically, it outperforms or matches existing algorithms on many atari games.

**Strengths:**

- The algorithm is clearly proposed and can be easily understood.
- The authors study both theoretical (mostly contraction property) and empirical aspects of the proposed algorithm.
- The experimental results seem comprehensive (on the entire atari games). They show that the algorithm performs better on complicated games.

**Weaknesses:**

After reading the paper I am still not sure about the benefits of using sinkhorn divergence, although I agree that the authors established some theoretical results (which I also have some concerns about below) and showed experimental results. My question is more intuitive---why should we pick Sinkhorn over others at a high level of intuition? I appreciate that the authors have spent many words explaining it from various perspectives. However, I didn't totally understand them and am not convinced at this moment. Some of my specific confusion are below:

- line 48-51: why does using samples instead of pre-specified statistics have better accuracy? Is there any reference?
- line 55: how is the regularization aligning with the max entropy principle? This point is mentioned multiple times in the paper but not explained in detail.
- line 53: "smoother" is very vague in this context. I would suggest using some different words.

As for the theoretical results, I am not sure how novel they are. For instance, the authors show convergence rate and sample complexity only under limit (when epsilon is approaching zero or infinity). Considering that the algorithm reduces to either QRDQN or MMD under the limit, the results are not so surprising and are actually quite natural. In addition, I didn't find general convergence results (I mean under any value of epsilon). Hence I am doubting if there is anything significant enough in the theory.

For the experiments, it generally looks good to me. I only have a minor comment for now: figure 2 seems a bit misleading--specifically, I feel plotting the ratio "Sinkhorn/QRDQN" is misleading. Think about two cases: if Sinkhorn is twice QRDQN, then the value will be 200%; on the other hand if QRDQN is twice Sinkhorn, then the ratio will be 50%. The two cases are symmetric but 200% will be visually four times higher than 50% in the plot. This may be why the Sinkhorn visually has a huge advantage over others in the figure.

**Questions:**

Line 319: the algorithm is much better on complicated games, which is very interesting. In other words, the algorithm is probably inferior on easy games. Do you have more explanation on this?

**Limitations:**

I didn't find any potential negative societal impact.

---

> ### Author Rebuttal · Authors · 2024-08-03
>
> Thank you for taking the time to review our paper. We appreciate your positive assessment and insightful feedback, and we would like to address the concerns you raised in your review.
>
>
> >My question is more intuitive---why should we pick Sinkhorn over others at a high level of intuition?
>
> We summarize the explanations in the introduction to provide a high-level intuition. Exiting distributional RL algorithms that rely on Wasserstein distance often struggle with 1) inaccuracies in representing distribution using pre-specified statistics, i.e., quantiles, and 2) difficulties in extending to the multi-dimensional reward setting. **Sinkhorn divergence stands out for efficiently approximating multi-dimensional Wasserstein distance** (by introducing an entropic regularization), simultaneously overcoming the two limitations. Additionally, the smoother transport plan and induced smoothness by the entropic regularization can also benefit the optimization in the RL learning process, making it a preferred choice.
>
>
> >line 48-51: why does using samples instead of pre-specified statistics have better accuracy? Is there any reference?
>
> Yes, using samples provides a more direct and accurate representation of return distributions. The MMD-DQN paper [1] discussed this extensively, and a more recent study [2] also adopts sample-based representation. These approaches, rooted in kernel methods, contrast with our method's foundation in optimal transport, offering a different approach to compare the distribution difference.
>
> >line 55: how is the regularization aligning with the max entropy principle? This point is mentioned multiple times in the paper but not explained in detail.
>
> According to the Sinkhorn divergence literature, the KL regularization in essence **amounts to vanilla entropy regularization** in terms of the joint distribution / optimal coupling. This is because:
> $$\begin{aligned}
> \text{KL}(\Pi|\mu \otimes \nu) = \mathcal{H}(\mu) + \mathcal{H}(\nu) - \mathcal{H}(\Pi).
> \end{aligned}
> $$
> where $\mathcal{H}$ is the entropy. Thus, the objective function is equivalent to
> $$\begin{aligned}
>  \min _{\Pi \in \mathbf{\Pi}(\mu, \nu)} \int c(x, y) \mathrm{d} \Pi(x, y) + \varepsilon \text{KL}(\Pi|\mu \otimes \nu) \iff  \min _{\Pi \in \mathbf{\Pi}(\mu, \nu)} \int c(x, y) \mathrm{d} \Pi(x, y) - \varepsilon \mathcal{H}(\Pi)
> \end{aligned}
> $$
> where the $\mathcal{H}(\mu)$ and $\mathcal{H}(\nu)$ are constant for fixed marginal distributions, which are independent of the minimizer. In this equivalent objective function form, minimizing the Sinkhorn divergence also encourages to maximize the entropy $\mathcal{H}(\Pi)$, leading to a more uniformly distributed optimal transport plan / joint distribution. This aligns well with the maximum entropy principle.
>
>
> >line 53: "smoother" is very vague in this context. I would suggest using some different words.
>
> Thanks for bringing this to our attention. The term "smoother" refers to the effect of the regularization in Sinkhorn divergence, which **encourages a more uniformly distributed transport plan**, as analyzed above. This more uniformly distributed transport plan is in contrast to the potentially sparse plan resulting from optimizing the unregularized Wasserstein distance. We will refine this terminology and provide a clearer explanation in the revised version.
>
> >As for the theoretical results, I am not sure how novel they are. ... the results are not so surprising and are actually quite natural. In addition, I didn't find general convergence results (I mean under any value of epsilon) ...
>
> We want to clarify that **Theorem 1(3) is precisely the contraction conclusion under the general $\epsilon \in (0, +\infty)$**, which is the main theoretical contribution of this paper. We encourage a further review of Theorem 1(3) and the subsequent proof sketch on Page 5 of our paper.
>
> >figure 2 seems a bit misleading--specifically, I feel plotting the ratio "Sinkhorn/QRDQN" is misleading. Think about two cases: if Sinkhorn is twice QRDQN, then the value will be 200%; on the other hand if QRDQN is twice Sinkhorn, then the ratio will be 50%. The two cases are symmetric but 200% will be visually four times higher than 50% in the plot. This may be why the Sinkhorn visually has a huge advantage over others in the figure.
>
> We acknowledge the **asymmetry** of the ratio improvement/percentage increase metric, i.e., $(A-B)/B$, in the visual representation, but it is intuitively interpretable and thus more commonly used in practice.
>
> Another metric is the log difference metric, i.e., $\log(A) - \log(B)$, which is symmetric as $\log(A/B)=\log(k)$ vs $\log(A/B)=\log(1/k)=-\log(k)$. However, the log differences are undefined for zero or negative values, which is its main limitation. As **the raw score in each Atari game can be either positive or negative**, the ratio improvement/percentage increase metric is more broadly applicable and particularly preferable in evaluating the RL algorithm across a wide range of Atari games than other metrics like the log difference.
>
> >Line 319: the algorithm is much better on complicated games, which is very interesting. In other words, the algorithm is probably inferior on easy games. Do you have more explanation on this?
>
> The finding that the algorithm is much better on complicated games does not imply that it is inferior on easy games. After checking the raw score table across 55 Atari games, we find our algorithm is also on par with or even performs better than other baselines on easy games, such as Atlantis, Enduro, FishingDerby, and MsPacman.
>
> We appreciate your insights and suggestions, which will guide our revisions to improve the clarity and accuracy of our manuscript. Please feel free to let us know if you have any further questions.
>
> ## Reference
>
> [1] Distributional Reinforcement Learning via Moment Matching (AAAI 2021)
>
> [2] Distributional Bellman Operators over Mean Embeddings (ICML 2024)

---

> > ### Comment · Reviewer_uu9w · 2024-08-11
> >
> > Thank you for the detailed response! It looks good to me, so I will maintain my positive score.

---

> ### Comment · Area_Chair_wJXb · 2024-08-11
> **Please respond to authors**
>
> Hello Reviewer uu9w: The authors have responded to your comments. I would expect that you would respond in kind.

---

### Official Review · Reviewer_Fdqh · 2024-07-12

**Soundness:** 3
**Presentation:** 2
**Contribution:** 3
**Rating:** 5
**Confidence:** 5

**Summary:**

•	This paper proposes a novel distributional RL algorithm, called SinkhornDRL, which interpolates between Wasserstein distance and MMD. The authors aim to estimate the distribution using unrestricted statistics, enhancing stability and facilitating extension to multi-dimensional reward settings. The authors also provide some theoretical guarantees on convergence of SinkhornDRL. The proposed method shows decent experimental performance in 55 Atari games and several multi-dimensional reward settings.

**Strengths:**

•	The paper reviews the relevant literatures in distributional RL and analyzes the new Sinkhorn divergence metric as an alternative to Wasserstein loss. The author provides sufficient background and advantages of Sinkhorn divergence, demonstrating a well-motivated approach.

•	Although the evaluation scores are presented for 40 million frames rather than 200 million, the authors ensure transparency and reproducibility by reporting raw scores for performance comparisons. The authors also present a significant amount of experiment and ablation studies to provide sufficient experimental results for the effectiveness of their algorithm.

**Weaknesses:**

•	Typos: In line 52, Sinkrhorn -> Sinkhorn

•	The text and figures in the paper are quite dense and difficult to read. Some parts seem unnecessary to be included in the main paper.

o	Algorithm 1 lacks certain technical details, and may rather be placed in Appendix. In addition, explaining the details of Algorithm 2 on Line 769 would help a lot in implementing a practical algorithm.

o	The text in Figure 2 is not legible. It would be better to include a table showing the mean and median of the best HNS scores, as commonly done in other distributional RL papers [1,2].

•	There may be a technical error regarding Theorem 1 (3). The authors state in Eq (33) that a universal upper bound $\bar{\Delta}_{\epsilon}(a,\alpha)$ is strictly less than 1. But if $\mu$ and $\nu$ are close enough, even if non-trivial, doesn’t the scaling factor become 1 by taking the supremum? In Line 660, the authors recognize the non-expansion case, but it seems to rely on the strong assumption that the set $\{\lambda_{\epsilon}(U,V)\}$ is finite. This assumption essentially implies that the set $\{\lambda_{\epsilon}(\(mathcal{T}^{\pi}_D)^n U, (\mathcal{T}^{\pi}_D)^n V )\}$ where $n \in \mathbb{N}$ is finite.

[1] Dabney, W., Rowland, M., Bellemare, M. G. & Munos, R. Distributional Reinforcement Learning with Quantile Regression. Arxiv (2017).

[2] Hessel, M. et al. Rainbow: Combining Improvements in Deep Reinforcement Learning. arXiv (2017) doi:10.48550/arxiv.1710.02298.

**Questions:**

•	Could the author elaborate on the statement in Lines 129-134 that 'SinkhornDRL inherently captures the spatial and geometric layout of return distributions'? Does this imply that the KL divergence term can leverage a richer representation of data geometry?

•	The definition of Sinkhorn divergence for multi-dimensional reward settings is unclear. Specifically, in Line 756, the cost function applies equal weights among reward sources. Can this be generalized to a weighted sum?

**Limitations:**

•	In line 22, the reference [11] is cited as a risk-sensitive control, but this is close to exploration method for distributional RL.

---

> ### Author Rebuttal · Authors · 2024-08-03
>
> Thank you for taking the time to review our paper. We appreciate your positive assessment and insightful feedback and would like to address the concerns you raised in the Weakness and Question parts of your review.
>
> >Weakness 1: The text and figures in the paper are quite dense and difficult to read. Some parts seem unnecessary to be included in the main paper ....
>
> * We acknowledge that Algorithm 1 does not involve too many technical details as it is aimed at outlining a generic update procedure of SinkhornDRL. As mentioned in Line 264, we have also provided a full version of SinkhornDRL in Algorithm 3 of Appendix G, in which we add more necessary technical details. In terms of Algorithm 2, given that Sinkhorn iteration is a well-established algorithm with guaranteed convergence, we thus leave it in the Appendix. We appreciate your suggestion and will add more explanations about Algorithm 2 in the appendix.
>
> * Thanks for this great advice. Following your suggestion, we have also **included a table to summarize the results in the global response**. It shows that our proposed SinkhornDRL achieves the best performance in terms of Mean and IQM(5%) HNS as well as the "> DQN" metric, and is on par with MMD-DQN in terms of Median of HNS.
>
> >Weakness 2: There may be a technical error regarding Theorem 1 (3). ...uniform upper bound and finite set assumption.
>
> The contraction conclusion indeed requires a uniform upper bound (also after taking the supremum) over all possible return distributions of $\Delta_{\epsilon}^{U, V}$ to remain strictly below 1, which can be satisfied by straightforwardly assuming a finite (**yet arbitrarily large** in the state and action spaces) MDP. The finite MDP ensures a finite return distribution set, further implying a finite set of $\lambda_{\epsilon}(U, V) \in (0, 1)$. Consequently, it is straightforward to establish that $\sup_{U, V} \Delta_{\epsilon}^{U, V} < 1$ as shown in Eq.21 under this finite condition. We argue that the finite MDP is not a strong condition but a common setting as most RL theory is established on it. In addition, we can also relax this finite MDP assumption by ruling out the extreme case, where there exists a series of elements in this set that can be arbitrarily close to 1. We have provided an example of this extreme case in Line 666. Since expressing this extreme case mathematically is complicated, we, therefore, retain the more common (and mild) finite MDP condition.
>
>
> >Question 1: Could the author elaborate on the statement in Lines 129-134 that 'SinkhornDRL inherently captures the spatial and geometric layout of return distributions'? Does this imply that the KL divergence term can leverage a richer representation of data geometry?
>
> By the nature of optimal transport distances, Sinkhorn divergence inherently captures the spatial and geometric layout, which can be demonstrated by its definition. In particular, Sinkhorn divergence and other optimal transports are defined by measuring the cost of moving mass from one point to another in a space, where the cost directly depends on the spatial distances between points. This is because the optimal transport plan is not just about matching quantiles but also **considers where mass is located and where it needs to go**, which inherently reflects the spatial relationships and geometric layout of distributions. This is also the primary advantage of optimal transport distances over others, e.g., MMD. In addition, the KL divergence also involves the joint distribution (transport plan), which retains the properties of optimal transport distances.
>
>
> >Question 2: The definition of Sinkhorn divergence for multi-dimensional reward settings is unclear. Specifically, in Line 756, the cost function applies equal weights among reward sources. Can this be generalized to a weighted sum?
>
> Yes, it is feasible to generalize the cost function to a weighted form, akin to an extension from Euclidean distance to Mahalanobis distance. By using the weighted cost function, the optimal transport distances can be **more closely aligned with specific applications.** The weighted cost function can potentially lead to benefits, provided that we have access to accurate or relevant weights, which typically require prior knowledge.
>
> >Limitation:  In line 22, the reference [11] is cited as a risk-sensitive control, but this is close to the exploration method for distributional RL.
>
> We acknowledge this citation issue and will revise it accordingly. Thanks for bringing this to our attention.
>
> We appreciate your insights and suggestions, which will guide our revisions to improve the clarity and accuracy of our manuscript. Please feel free to let us know if you have any further questions.

---

> ### Comment · Area_Chair_wJXb · 2024-08-11
> **Reviewer Fdqh please respond**
>
> Hello Reviewer Fdqh: The authors have responded to your comments. I would expect that you would respond in kind.

---

### Official Review · Reviewer_nXJK · 2024-07-12

**Soundness:** 3
**Presentation:** 4
**Contribution:** 3
**Rating:** 7
**Confidence:** 3

**Summary:**

This paper introduces Sinkhorn Distributional Reinforcement Learning (SinkhornDRL), a new algorithm designed to address the limitations of current distributional RL methods, particularly those relying on quantile regression. Existing methods often struggle with accurately capturing the characteristics of return distributions and extending to scenarios with multi-dimensional rewards. SinkhornDRL leverages Sinkhorn divergence, a regularized Wasserstein loss, to minimize the difference between current and target Bellman return distributions. This approach combines the geometric advantages of Wasserstein distance with the computational efficiency of Maximum Mean Discrepancy (MMD). The paper provides theoretical proof of SinkhornDRL's contraction properties, demonstrating its convergence behavior and relationship to other distance metrics. Empirical evaluations on the Atari games suite show that SinkhornDRL consistently outperforms or matches existing algorithms, especially in settings with multi-dimensional rewards.

**Strengths:**

- Introduces a new family of distributional RL algorithms based on Sinkhorn divergence, expanding the toolkit for researchers and practitioners.

- Provides theoretical analysis of Sinkhorn divergence in the context of distributional RL, including convergence guarantees.

- Demonstrates the effectiveness of SinkhornDRL through extensive experiments on a standard benchmark, showing superior performance in many cases.

- Specifically tackles the issues of inaccurate distribution capture and difficulty with multi-dimensional rewards that plague quantile regression methods.

**Weaknesses:**

- SinkhornDRL introduces some additional computational cost compared to simpler methods like C51 and QR-DQN.

- The algorithm requires tuning of additional hyperparameters (e.g., the regularization strength), which might require extra effort in practice.

- The paper acknowledges that a deeper connection between theoretical properties of divergences and practical performance in specific environments remains an open question.

- Some typos:
  - line 151 and line 647 Sinkrhon
  - line 163 supremal from
  - line 787 numeral

**Questions:**

Could the authors address the following points:

- Are there existing works on distributional reinforcement learning that use the entropic regularized Wasserstein distance?
- What specific advantages does the Sinkhorn divergence offer over the entropic regularized Wasserstein distance in the context of this study?

---

> ### Author Rebuttal · Authors · 2024-08-03
>
> Thank you for taking the time to review our paper. We appreciate your positive assessment and insightful feedback and would like to address the concerns you raised in your review.
>
>
> >Question 1: Are there existing works on distributional reinforcement learning that use the entropic regularized Wasserstein distance?
>
> To our knowledge, our study is the first to investigate the entropic regularized Wasserstein distance in the context of distributional RL.
>
> >Question 2: What specific advantages does the Sinkhorn divergence offer over the entropic regularized Wasserstein distance in the context of this study?
>
> As highlighted in our introduction, the specific advantages of Sinkhorn divergence, which is an entropic regularized Wasserstein distance, over the vanilla Wasserstein distance in distributional RL can be summarized as follows.
>
> * **Enhanced accuracy in representing return distributions** Sinkhorn divergence uses the samples to depict return distributions, offering more flexibility than the pre-specified statistics, e.g., quantiles, used in vanilla Wasserstein distance. Consequently, our approach naturally circumvents the non-crossing issues of the learned quantile curves.
>
> * **Applicability to multi-dimensional rewards** Many RL tasks involve multi-dimensional reward structures, but it remains elusive how to use quantile regressions to approximate a multi-dimensional Wasserstein distance. In contrast, Sinkhorn divergence can effectively approximate a multi-dimensional Wasserstein distance, thus providing an efficient solution to address these RL tasks,
>
> * **Increased robustness to noises via smoother transport plans** The entropic regularization incorporated in Sinkhorn divergence fosters smoother transport plans compared with those derived from unregularized Wasserstein distance. This makes it less sensitive to noises in the learning process.
>
> * **Stable optimization in RL learning** The entropic regularization turns the objective into a strongly convex problem, and the induced smoothness facilitates faster and more stable convergence in RL learning.
>
> We acknowledge the typos you pointed out and will correct them. Thank you once again for the time and effort you dedicated to reviewing our work.

---

> ### Comment · Reviewer_nXJK · 2024-08-10
>
> Thank you for your responses! It seems there was a misunderstanding. For both questions 1 and 2, I was referring to the entropic regularized Wasserstein distance W_{c, \varepsilon}. In the paper, you adopted the Sinkhorn divergence \bar{W}_{c, \varepsilon}. I was curious why the entropic regularized Wasserstein distance W_{c, \varepsilon} was not used instead.

---

> > ### Author Response · Authors · 2024-08-10
> >
> > We apologize for our misunderstanding. In practice, including in various applications where Sinkhorn divergence is used, the corrected entropic regularization Wasserstein distance $\overline{W}_ {c, \varepsilon}$ is typically preferred over the uncorrected one $W_{c, \varepsilon}$.
> >
> > As mentioned in Line 157 in our paper, $\overline{W}_ {c, \varepsilon}$ subtracts two self-distance terms to **correct the bias** introduced by the entropy regularization. This correction is necessary as $W_{c, \varepsilon}$ introduces a bias such that $W_{c, \varepsilon}(\mu, \mu) \neq 0$ for any $\mu$, which is also discussed in  [1] and [2]. Additionally, subtracting the two self-distances also ensures **non-negativity and metric properties**.
> >
> > On the contrary, directly using the uncorrected $W_{c, \varepsilon}$ will introduce bias when evaluating the distance between the current and target return distributions during the optimization of distributional RL, thus undermining the algorithm's performance. Therefore, it is essential to leverage the corrected version $\overline{W}_ {c, \varepsilon}$  in real applications to **ensure accurate and unbiased distance measurements**.
> >
> > ### Reference
> >
> > [1] Feydy, Jean, et al. "Interpolating between optimal transport and mmd using sinkhorn divergences." (AISTATS 2019).
> >
> > [2] Genevay, Aude, et al. "Sample complexity of Sinkhorn divergences." (AISTATS 2019)

---

> > > ### Comment · Reviewer_nXJK · 2024-08-10
> > >
> > > Thanks! That makes a lot of sense.

---

### Author Rebuttal · Authors · 2024-08-03

Dear Reviewers,

We would like to thank all the reviewers for their thoughtful and constructive feedback on our paper. We deeply appreciate your positive assessment and have thoroughly provided our response to address each of your concerns.  We are committed to enhancing the quality of our work and remain at your disposal for any further clarifications or questions you might have.

(Attached is the Summary Table result in response to Reviewer Fdqh)

Sincerely,
Authors

---

### Decision · Program_Chairs · 2024-09-25

**Decision:**

Accept (poster)

**Comment:**

Distributional RL approximates the full distribution of returns, which contrasts with classic expected-value RL. Quantiles are the method of choice for quantifying the distribution of returns, but are limited in their expressiveness. The paper proposes Sinkhorn distributional RL to address limitations in expressiveness. Theoretically, the paper presents a contraction result and empirically the results show improvements over previous Distributional RL results over 40M training frames.

The reviewers are uniformly positive about the paper. I have read the paper and agree that the idea is interesting and will be appreciated by the community.